# The critical role of Hedgehog-responsive mesenchymal progenitors in meniscus development and injury repair

Yulong Wei[1,2†], Hao Sun[1,3†], Tao Gui[1,4†], Lutian Yao[1], Leilei Zhong[1], Wei Yu[1,2], Su-Jin Heo[1,5], Lin Han[6], Nathaniel A Dyment[1], Xiaowei Sherry Liu[1], Yejia Zhang[1,5,7], Eiki Koyama[8], Fanxin Long[8], Miltiadis H Zgonis[1], Robert L Mauck[1,5], Jaimo Ahn[1,9], Ling Qin[1]*

[1]Department of Orthopaedic Surgery, Perelman School of Medicine, University of Pennsylvania, Philadelphia, United States; [2]Department of Orthopaedics, Union Hospital, Tongji Medical College, Huazhong University of Science and Technology, Wuhan, China; [3]Department of Orthopedics, Sun Yat-Sen Memorial Hospital, Sun Yat-Sen University, Guangzhou, China; [4]Department of Bone and Joint Surgery, Institute of Orthopedic Diseases, The First Affiliated Hospital, Jinan University, Guangzhou, China; [5]Translational Musculoskeletal Research Center, Corporal Michael J. Crescenz VA Medical Center, Philadelphia, United States; [6]School of Biomedical Engineering, Science and Health Systems, Drexel University, Philadelphia, United States; [7]Department of Physical Medicine and Rehabilitation, Perelman School of Medicine, University of Pennsylvania, Philadelphia, United States; [8]Translational Research Program in Pediatric Orthopaedics, The Children's Hospital of Philadelphia, Philadelphia, United States; [9]Department of Orthopaedic Surgery, University of Michigan Medical School, Ann Arbor, United States

*For correspondence:
qinling@pennmedicine.upenn.edu

†These authors contributed equally to this work

Competing interests: The authors declare that no competing interests exist.

**Abstract** Meniscal tears are associated with a high risk of osteoarthritis but currently have no disease-modifying therapies. Using a Gli1 reporter line, we found that Gli1[+] cells contribute to the development of meniscus horns from 2 weeks of age. In adult mice, Gli1[+] cells resided at the superficial layer of meniscus and expressed known mesenchymal progenitor markers. In culture, meniscal Gli1[+] cells possessed high progenitor activities under the control of Hh signal. Meniscus injury at the anterior horn induced a quick expansion of *Gli1-lineage* cells. Normally, meniscal tissue healed slowly, leading to cartilage degeneration. Ablation of Gli1[+] cells further hindered this repair process. Strikingly, intra-articular injection of Gli1[+] meniscal cells or an Hh agonist right after injury accelerated the bridging of the interrupted ends and attenuated signs of osteoarthritis. Taken together, our work identified a novel progenitor population in meniscus and proposes a new treatment for repairing injured meniscus and preventing osteoarthritis.

## Introduction

Meniscal tears, with all the morbidity and disability they cause, are among the most common injuries of the knee affecting both the young and aged; and the procedures to address them are among the most commonly performed surgeries in the orthopedics field. Beyond the short-term pain, disability, time from desired activities including work, meniscal injuries are important early events in the initiation and later propagation of osteoarthritis (OA) (*Lohmander et al., 2007*). From a clinical therapeutic point of view, surgical treatments, including the maximally preserving partial meniscectomy, while improving immediate symptoms, do not delay the natural history progression of OA or may actually

accelerate it. As the adult meniscus is predominantly avascular, true biologic healing with surgical repair remains a viable treatment for only a small portion of individuals typically with tears contained within the red vascular zone (*Makris et al., 2011*). For the majority of the injuries, a restorative bio-logic therapy does not currently exist in practice.

Mesenchymal progenitors play a critical role in tissue regeneration. Therefore, identifying and characterizing residential mesenchymal progenitors in meniscus are important for developing novel and effective strategies to treat meniscus injury. Using enzymatic digestion and clonal expansion methods, previous studies have demonstrated that human and rabbit meniscus contain mesenchy-mal progenitors with multi-differentiation abilities (*Gui et al., 2015*; *Huang et al., 2016*; *Segawa et al., 2009*; *Shen et al., 2014*). Interestingly, the superficial layer of meniscus was pro-posed to harbor the progenitors. By collecting cells growing out of mouse meniscus explant, Gamer et al. showed that these cells exhibit stem cell-like characteristics and are located in the superficial zone in vivo (*Gamer et al., 2017a*). During injury, it has been observed that progenitors on the meniscus surface migrate from vascularized red zone to non-vascularized white zone for repair (*Seol et al., 2017*). While these cells in culture express several common mesenchymal progenitor markers, such as CD44, Sca1, and CD90, their in vivo properties and regulatory signaling pathways are not known (*Shen et al., 2014*; *Seol et al., 2017*).

Hedgehog (Hh) signaling is essential for embryonic development and tissue homeostasis. It is one of few fundamental pathways that maintain adult stem and progenitor cells in various organs, such as brain, skin, bladder, teeth, and others (*Petrova and Joyner, 2014*). Following injury, Hh signaling can trigger stem and other resident cells to participate in repair, and therefore, Hh upregulation is viewed not only as a natural response to injury but also as a way to stimulate tissue repair by activat-ing stem cells. Gli1, an integral effector protein of Hh pathway, was recently recognized as a marker for bone marrow, periosteal, and periarticular mesenchymal progenitors (*Shi et al., 2017*; *Tong et al., 2019*; *Wang et al., 2019*), suggesting that Hh signaling is also functional in the skeleton for maintaining tissue-specific stem and progenitors.

In this study, we constructed a Gli1 reporter line and found that Gli1-labeled Td$^+$ cells are exclu-sively located in the horns of adolescent meniscus. These cells contribute to meniscus development and possess mesenchymal progenitor properties. In adult mice, Gli1$^+$ cells mostly reside at the superficial layer of meniscus and they rarely become cells in the center of meniscus. Interestingly, meniscus injury induced a rapid expansion of Gli1-lineage cells and elimination of these cells miti-gated repair. Using sorted Gli1$^+$ cells and an Hh signaling agonist, we demonstrated that activating Hh signaling could be an effective way to promote meniscus repair and prevent OA progression.

## Results

### The expression patterns of Gli1$^+$ cells and their descendants in mouse meniscus

We performed lineage tracing with *Gli1$^{CreER}$ Rosa26 $^{lsl-tdTomato}$* (Gli1ER/Td) mice at various ages to identify Gli1$^+$ cells and their descendants in the meniscus following a 6 week chase (*Figure 1A*). Joints were cut in either sagittal or coronal planes to visualize different parts of meniscus (*Figure 1B*). In line with our previous report (*Tong et al., 2019*), at 1 week of age, Gli1$^+$ cells were only observed in the periarticular layer of articular cartilage, but not in the meniscus and other joint tissues (*Figure 1Ca-c*). Long term tracing also did not detect any Td signal in the meniscus, confirm-ing that neonatal meniscus does not harbor Gli1$^+$ cells (*Figure 1Cd*). At 2 weeks of age, most cells in the anterior horn of the meniscus, both medially and laterally, were Td$^+$ (*Figure 1Ce-g*). Six weeks of tracing confirmed that the entire anterior horn, but not the posterior horn, is labeled by Td (*Figure 1Ch*). At 4 weeks of age, Gli1$^+$ cells were concentrated in the superficial layer of the anterior horn; 6 weeks later, most cells in both superficial and central portions of the anterior horn were labeled by Td (*Figure 1Ci–l*). Within the posterior horn, very few cells in the center of meniscus were initially labeled but then gave rise to the majority of internal cells 6 weeks later. Quantification along the length of the meniscus over the time indicated that 1–8 weeks of age represents the rapid grow-ing phase for the meniscus (*Figure 1—figure supplement 1*). Taken together, our data suggested that Gli1$^+$ cells represent progenitors for meniscus cells of the horn regions at adolescence stage.

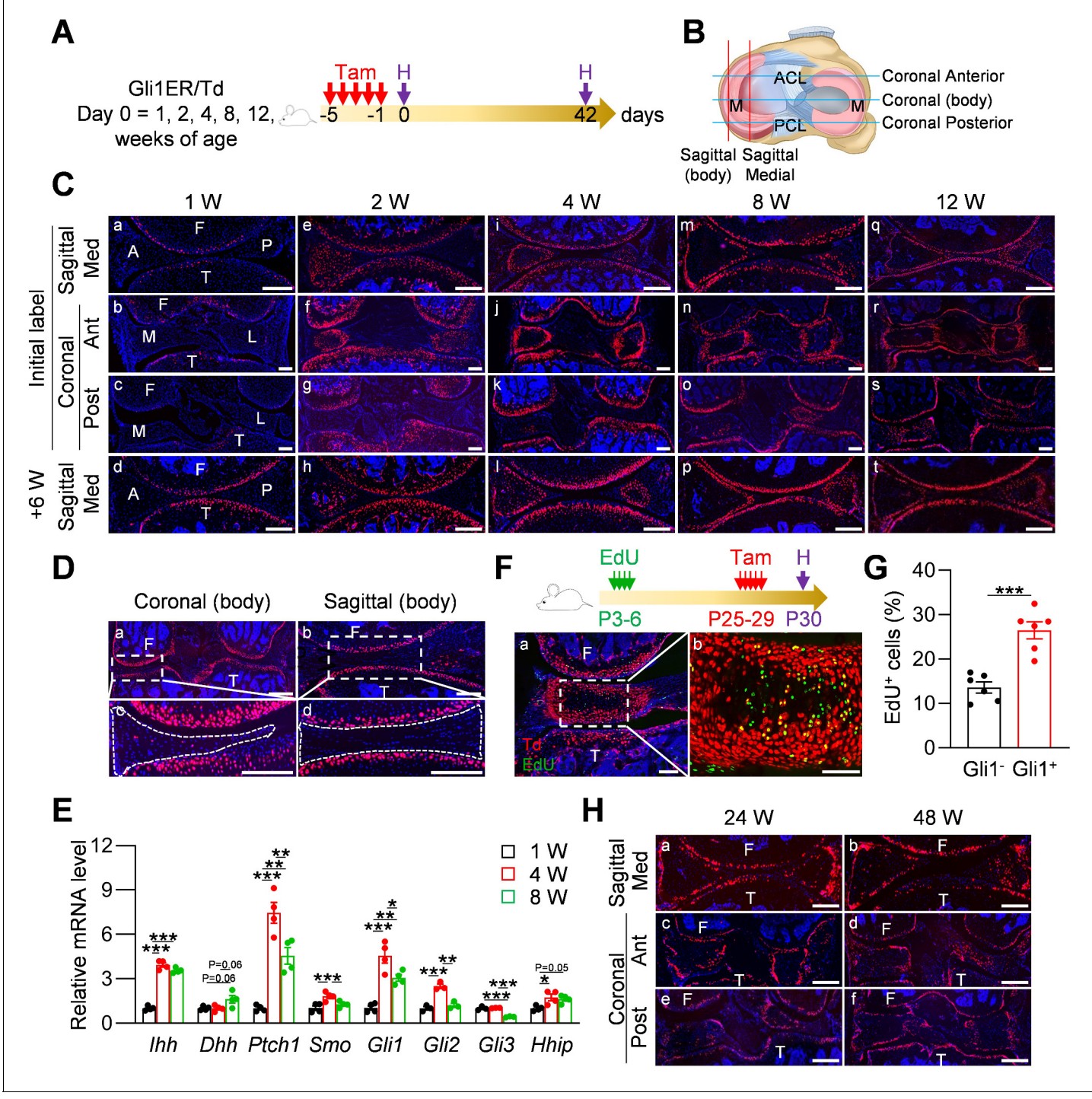

**Figure 1.** Gli1 labels mesenchymal progenitors in mouse meniscus during development. (**A**) Schematic graph of the study protocol. Male Gli1ER/Td mice were treated with Tam at 1, 2, 4, 8, and 12 weeks of age and analyzed at 24 hr (pulse) or 6 weeks (tracing) after the last Tam dosing. (**B**) Schematic cartoon of meniscus shows sectioning sites. M: meniscus; ACL: anterior cruciate ligament; PCL: posterior cruciate ligament. (**C**) Representative fluorescence images of meniscus sections at indicated ages and sectioning sites. n = 3 mice/age/sectioning site. Scale bars, 200 μm. F: femur; T: tibia; A: anterior; P: posterior; M: medial meniscus; L: lateral meniscus; Med: medial; Ant: anterior; Post: posterior. Red: Td; Blue: DAPI. (**D**) Representative fluorescence images of meniscus body at coronal (**a**) and sagittal (**b**) planes from 12-week-old Gli1ER/Td mice. Meniscus were harvested at 24 hr after the last Tam injection. n = 3 mice/sectioning site. Scale bars, 200 μm. Boxed areas in a and b are shown at high magnification as c and d, respectively. Dashed lines outline meniscus. (**E**) qRT-PCR analysis of Hh signaling component genes in mouse meniscus tissues at 1, 4, 8 weeks of age. n = 4 independent experiments. (**F**) Top panel is a schematic representation of the study protocol. Gli1ER/Td mice were injected with EdU at P3-6 and Tam at P25-29. Joints were harvested 24 hr later. Representative confocal images of coronal sections of mouse knee joints are presented at the bottom

*Figure 1 continued on next page*

*Figure 1 continued*

panel. Boxed area in a (Scale bars, 200 µm) is shown at high magnification in b (Scale bars, 50 µm). Green: EdU. (**G**) The percentage of EdU$^+$ cells within Gli1$^+$ or Gli1$^-$ meniscus cells was quantified. n = 6 mice/group. (**H**) Gli1ER/Td mice were treated with Tam at 24 or 48 weeks of age and analyzed 24 hr later. Representative fluorescence images of sagittal (**a, b**) and coronal (**c–f**) sections of knee joints are presented. Scale bars, 200 µm. Statistical analysis was performed using unpaired two-tailed t-test and one-way ANOVA with Tukey-Kramer post-hoc test. Data presented as mean ± s.e.m. *p<0.05, **p<0.01, ***p<0.001.

The online version of this article includes the following source data and figure supplement(s) for figure 1:

**Source data 1.** Raw data for *Figure 1G and E*.
**Figure supplement 1.** Mouse meniscal morphogenesis during development.
**Figure supplement 2.** The density of Gli1$^+$ cells along meniscus surface was measured in mice at different ages.
**Figure supplement 3.** Meniscal enthesis and ligamental enthesis regions in joint are enriched with Gli1$^+$ cells.
**Figure supplement 4.** Gli1 labels superficial zone cells of mouse and mini-pig meniscal horns.

Starting from 8 weeks of age, Gli1$^+$ cells were exclusively restricted to the superficial layer of the anterior horn throughout tracing (*Figure 1* Cm-t). The labeling pattern in the posterior horn was slightly different that Td$^+$ cells first appear in the center and then expand to the entire tissue 6 weeks later (*Figure 1* Cm-p). At 12 weeks of age, Gli1$^+$ cells remained restricted to the superficial layer of both anterior and posterior horns throughout tracing (*Figure 1* Cq-t). At any given age, Td signal was not detected in the center of the body of either the medial or lateral meniscus regardless of cutting planes (*Figure 1D*).

To investigate whether the appearance of Gli1$^+$ cells are indeed correlated with enhanced Hh signaling, we analyzed the expression of Hh signaling components in mouse meniscus during postnatal development (*Figure 1E*). Interestingly, Ihh expression was greatly increased at 4 weeks of age and maintained at the elevated level at 8 weeks of age, while Dhh expression only showed a tendency to increase during the same time period. We also observed significance increases of *Ptch1, Smo, Gli1, Gli2, and Hhip* mRNA at 4 weeks of age compared to 1 week of age. Most increases declined at 8 weeks of age. Meanwhile, Gli3 expression decreased at 8 weeks of age and *Shh* mRNA was not detected at all time points. Thus, our data validated the enhanced Hh signaling during postnatal meniscal development and pointed out Ihh as the major responsible ligand.

Slow cycling cells are considered quiescent stem cells (*Moore and Lyle, 2011*). Applying a label-retention method on neonatal mice (EdU injections at P3-6 and Tam injections at P25-29), we found that Gli1$^+$ cells at P30 contain much more EdU$^+$ cells than Gli1$^-$ cells (*Figure 1F,G*), indicating that meniscus stem cells are enriched in the Gli1$^+$ cell population.

When mice reached mature and late adult stages (24 and 48 weeks of age, respectively), Gli1 mostly marked the superficial layer of both anterior and posterior horn of the meniscus (*Figure 1H*). Quantification of cells along the surface of meniscal horns revealed a drastic reduction of Gli1$^+$ cells in aged mice compared to adolescent mice (*Figure 1—figure supplement 2*).

Meniscus is attached to neighboring bones via fibrocartilaginous entheses. We found that Gli1 labels these entheses between anteromedial, posteromedial, anterolateral, posterolateral meniscus and the tibial plateau or femur condyle (*Figure 1—figure supplement 3A*). In addition, Gli1 also labeled the osseous ligamentous junctions between the anterior cruciate ligament or posterior cruciate ligament and femur or tibia (*Figure 1—figure supplement 3B,C*).

The existence of Gli1-labeled cells on the meniscus surface was confirmed by Gli1 immunostaining (*Figure 1—figure supplement 4A*). Furthermore, analysis of porcine meniscus revealed a similar staining pattern. As shown in *Figure 1—figure supplement 4B*, Gli1$^+$ cells were located in the superficial layer, but not in the central part, of meniscus horn in the adult mini-pig.

## Gli1-expressing meniscus cells are a major subset of mesenchymal progenitors

We next investigated whether Gli1$^+$ meniscus cells possess mesenchymal progenitor properties. For these experiments, 3-month-old Gli1ER/Td mice received Tam injections for five consecutive days and euthanized 2 days after the last injection for immunostaining and cell isolation. Immunostaining of mesenchymal markers, such as Sca1, Cd90, Cd200, PDGFRα (*Boxall and Jones, 2012*), and Cd248 (*Naylor et al., 2012*; *Bagley et al., 2009*) revealed their co-staining with Td$^+$ signal in the superficial layer of the meniscus (*Figure 2Aa-e*). Prg4 is the lubricant highly synthesized by cartilage,

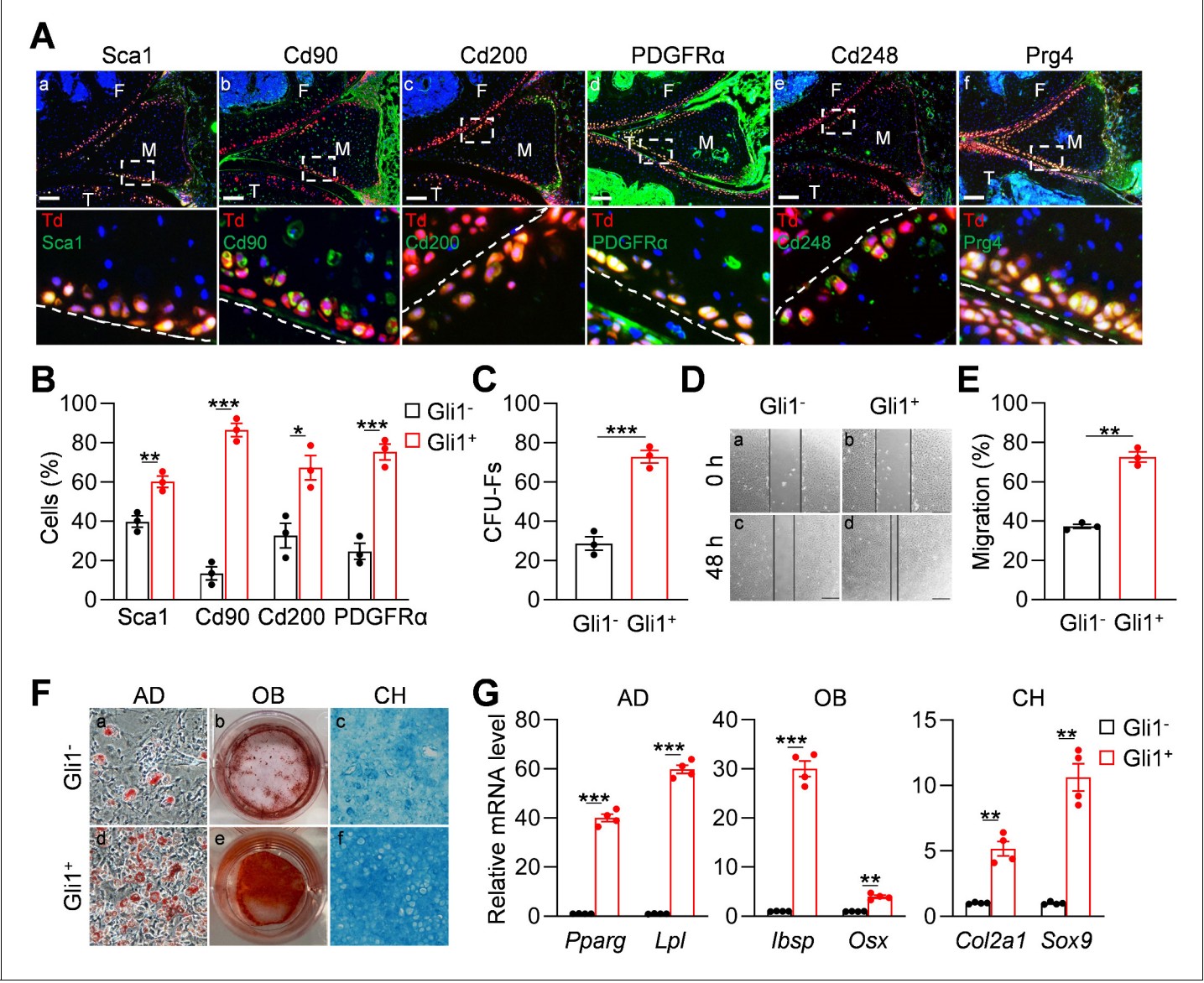

**Figure 2.** Gli1-labeled meniscus cells possess mesenchymal progenitor properties. (**A**) Representative immunofluorescence images of Sca1, Cd90, Cd200, PDGFRα, Cd248, and Prg4 staining in 3-month-old Gli1ER/Td meniscus. Scale bars, 200 μm. Boxed areas are shown at high magnification in corresponding panels at the bottom. Dashed lines indicate meniscus surface. Yellow cells are double positive for progenitor marker and Td. F: femur; T: tibia; M: meniscus. Blue: DAPI. (**B**) Quantification of the expression level of mesenchymal progenitor markers in Gli1+ and Gli1- cells from meniscus. Digested meniscus cells from 3-month-old Gli1ER/Td mice were subjected to flow cytometry analysis. n = 3 independent experiments. (**C**) CFU-F assay of digested meniscus cells. Td+colonies and Td- colonies were counted from 1 × 10^4 seeded cells. n = 3 independent experiments. (**D**) Representative bright-field images of the scratch-wound closure in Gli1+ or Gli1- meniscus cells at 0 and 48 hr. Scale bars, 200 μm. Solid lines indicate the remaining area not covered by meniscus cells. (**E**) The relative migration rate was measured by the percentage of scratched area being covered by migrated cells at 48 hr. n = 3 independent experiments. (**F**) Representative adipogenic (AD), osteogenic (OB), and chondrogenic (CH) differentiation images of Gli1+ and Gli1- cells. Cells were stained by Oil Red, Alizarin red, and Alcian blue, respectively. (**G**) qRT-PCR analysis of lineage markers in Gli1- or Gli1+meniscal cells after being cultured in adipogenic, osteogenic and chondrogenic differentiation media for 1, 2, and 3 weeks, respectively. n = 4 independent experiments. Statistical analysis was performed using unpaired two-tailed t-test. Data presented as mean ± s.e.m. *p<0.05, **p<0.01, ***p<0.001.

The online version of this article includes the following source data and figure supplement(s) for figure 2:

**Source data 1.** Raw data for *Figure 2B,C,E,G*.

**Figure supplement 1.** Mesenchymal progenitor markers are enriched in Gli1+ meniscus cells.

**Figure supplement 2.** Gli1+ cells proliferate faster than Gli1- cells.

meniscus and synovium surface cells (*Jia et al., 2016*; *Lee et al., 2008*). We found that Gli1[+] cells are also Prg4[+] (*Figure 2Af*). Using an enzymatic digestion approach, we harvested meniscus cells for subsequent studies. Flow cytometry revealed that Gli1[+] cells are only 2% of digested meniscus cells and that they express mesenchymal progenitor markers Sca1, CD90, CD200, and PDGFRα at a higher level than Gli1[-] cells (*Figure 2—figure supplement 1*, *Figure 2B*).

Cells digested from the entire meniscus formed CFU-Fs in dishes. Interestingly, 72.8% of CFU-Fs were Td[+], suggesting that Gli1[+] cells have a high clonogenic activity (*Figure 2C*). While both Td[+] and Td[-] cell can grow in culture, sorted Td[+] cells proliferated faster (*Figure 2—figure supplement 2A,B*) and migrated quicker (*Figure 2D,E*) than Td[-] cells. In addition, Td[+] meniscus cells differentiated better into osteoblasts, adipocytes, and chondrocytes than Td[-] cells as shown by staining

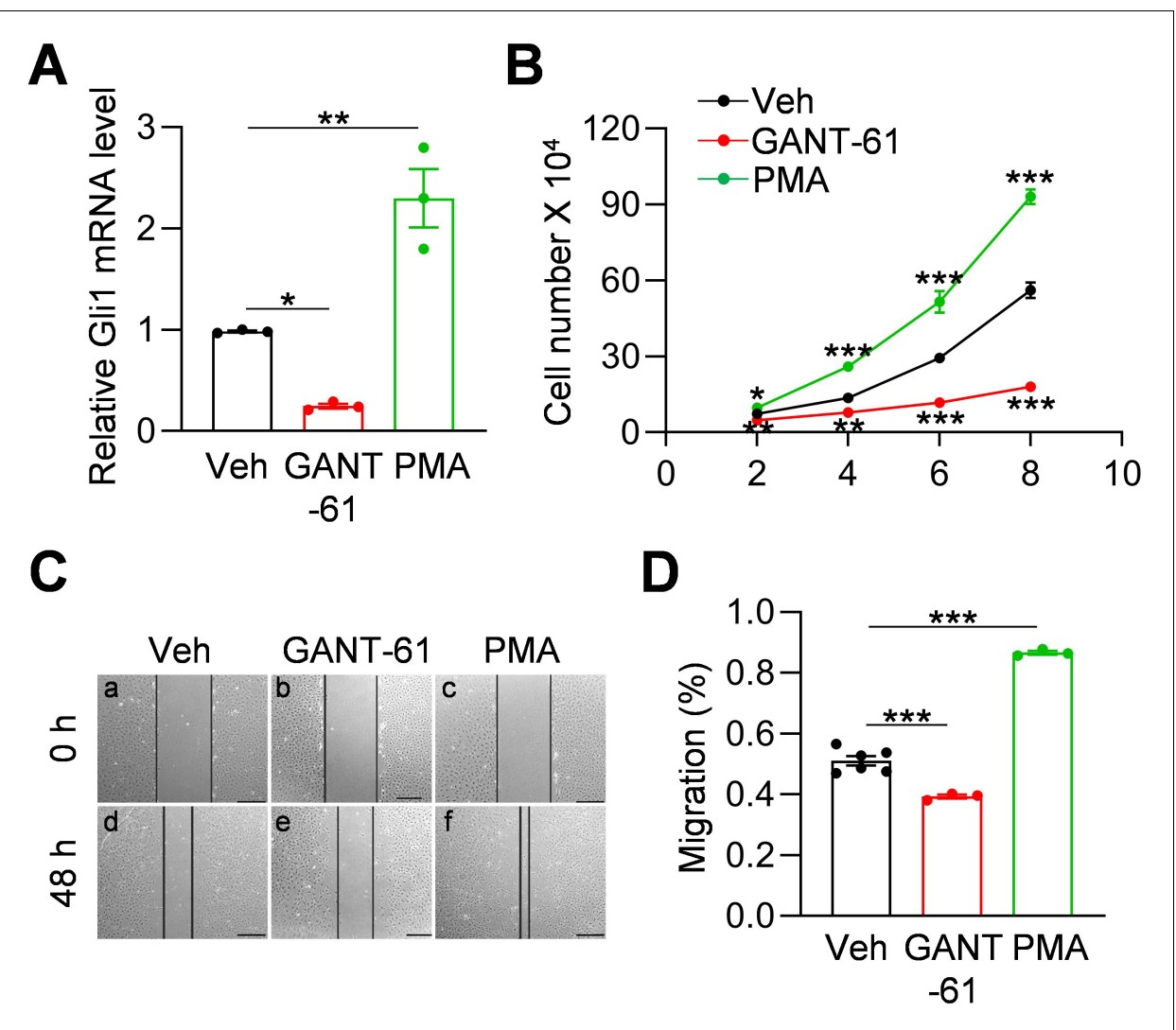

**Figure 3.** Hh signaling stimulates proliferation and migration of meniscus mesenchymal progenitors. (**A**) qRT-PCR analysis of Gli1 mRNA in primary mouse meniscus cells treated with vehicle, GANT-61 (10 µM) or PMA(1 µM) for 48 hr. n = 3 independent experiments. (**B**) The proliferative ability of primary mouse meniscus cells was up-regulated by PMA and down-regulated by GANT-61 over 8 days of culture. n = 3 independent experiments. (**C**) Representative bright-field images of the scratch-wound closure in meniscus cells treated with veh, GANT-61 or PMA after 48 hr. Scale bars, 200 µm. Solid lines indicate the remaining area not covered by meniscus cells. (**D**) The relative migration rate was measured. n = 3–6 independent experiments. Statistical analysis was performed using one-way ANOVA with Dunnett's post-hoc test. Data presented as mean ± s.e.m. *p<0.05, **p<0.01, ***p<0.001.

The online version of this article includes the following source data for figure 3:

**Source data 1.** Raw data for *Figure 3A,B,D*.

(*Figure 2F*) and marker gene expression (*Figure 2G*). Taken together, these data demonstrated that Gli1$^+$ cells possess the properties of mesenchymal progenitors: self-renewal and multi-lineage differentiation.

## Hh signaling regulates meniscus progenitors

Gli1 expression is a reporter for Hh signaling pathway (*Dagklis et al., 2016*). To investigate whether Hh signaling is involved in regulating meniscus progenitors, we treated meniscus progenitors with purmorphamine (PMA), an activator of Hh signaling (*Wu et al., 2004*), or Gli antagonist 61 (GANT-61), a Gli1 inhibitor (*Lauth et al., 2007*). In line with previous reports (*Srivastava et al., 2014*; *Wang et al., 2013*), GANT-61 reduced Gli1 expression in meniscus progenitors while PMA increased it (*Figure 3A*). Cell counting revealed that GANT-61 reduces the number of meniscus cells in culture over time while PMA increases cell number (*Figure 3B*). Similarly, scratch assay showed that Hh signaling promotes the migration of meniscus progenitors (*Figure 3C,D*). Our data demonstrated an important action of Hh signaling in promoting proliferation and migration of meniscus progenitors.

## Injury-induced Gli1-lineage cell expansion is critical for meniscus healing

Meniscal tear is a common injury in joints. To mimic this injury, we surgically cut the anteromedial horn of the meniscus into two parts in 3-month-old mice, resulting in disconnected synovial and ligamental ends of the meniscus (*Figure 4A*). Gli1ER/Td mice received Tam right before surgery (*Figure 4B*). At 1–2 weeks post-surgery, the two ends of the meniscus retracted toward the synovium and ligament, respectively (*Figure 4Ca-c*). This was accompanied by massive synovial hyperplasia that wrapped around the ends of the meniscus and likely stabilized them. At 4 weeks, the synovium returned to relatively normal thickness and the two cut ends of the meniscus were aligned but not connected (*Figure 4Cd*). Over time, the connection between the two ends gradually moved toward re-establishment but never reached the normal level even after 3 months post-surgery (*Figure 4Ce, f*). The meniscus repair scores summarized this trend (*Figure 4D*), suggesting that meniscus heals slowly in this injury model.

Fluorescence imaging was used to analyze the contribution of Gli1$^+$ cells and their descendants during this process. Strikingly, starting from 2 weeks post-injury, Td$^+$ cells appeared at the synovial ends and ligamental ends of injured meniscus (*Figure 4Cg-I*). Their number peaked around 4 weeks, and gradually declined thereafter (*Figure 4Cj-l*). Total cell density and the percentage of Td$^+$ cells at both ends were significantly increased after injury, particularly at the ligamental end (*Figure 4E,F*). qRT-PCR further demonstrated the increased expression of Hh signaling components after injury (*Figure 4G*). Interestingly, different from the development stage, Dhh, but not Ihh, was highly induced after injury, suggesting that Hh ligands play distinct roles in meniscal development and injury repair. An EdU incorporation experiment confirmed that many Gli1$^+$ cells and their progenies are proliferative at 2 weeks post surgery (*Figure 4H*). In old mice (52 weeks of age), this expansion of Td$^+$ cells after injury was remarkably attenuated and the end-to-end reconnection was much less with a lower repair score than young adult mice 4 weeks later (*Figure 4—figure supplement 1A,B*), indicating that aging diminishes the repair ability of meniscus.

To further understand the role of Gli1-lineage cell expansion in meniscus repair, we generated *Gli1$^{CreER}$ Rosa26$^{lsl-tdTomato}$ Rosa26$^{lsl-DTA}$* (Gli1ER/Td/DTA) mice for a cell ablation experiment. These mice at 3 months of age received Tam injections followed by meniscus injury (*Figure 4I*). One day after the Tam injections, Td$^+$ cells in meniscus were drastically decreased by 54.5%, as shown by both sagittal and coronal views of meniscus horns (*Figure 4J*). Three months later, while two meniscus ends loosely reconnected in vehicle-treated mice, those in Tam-treated mice were still well separated, leading to a significant reduction of repair score (*Figure 4K,L*). Fluorescence imaging confirmed no more expansion of Td$^+$ cells in Tam-treated mice. These data clearly indicate an essential role of Gli1-lineage cells in meniscal healing.

To validate our mouse data, we collected healthy and degenerated human meniscus for immunohistochemistry analysis. Degenerated meniscus had surface disruption, collagen fibers disorganization, and positive safranin O/fast green staining as previously reported (*Pauli et al., 2011*). Healthy meniscus did not show Gli1 staining (*Figure 4—figure supplement 2*). However, in moderate and severe degenerated meniscus, Gli1 was readily detectable in cell clusters formed in various sizes and characterized by Ki67$^+$ staining, indicating that Gli1$^+$ cells are proliferative. These results confirmed

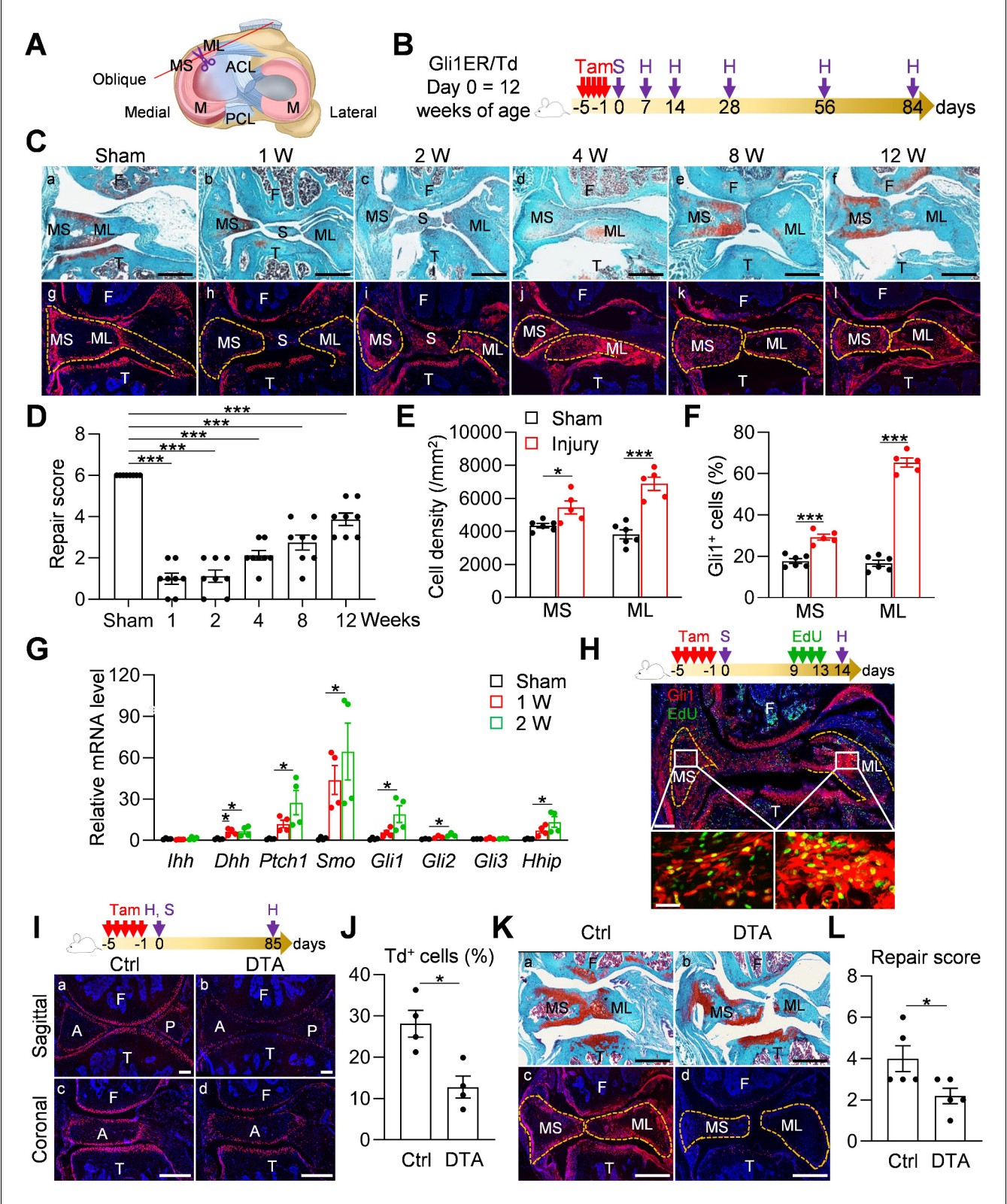

**Figure 4.** Meniscus injury rapidly expands Gli1-lineage cells. (**A**) Schematic cartoon of meniscus shows the sectioning site. M: meniscus; MS: meniscus synovial end; ML: meniscus ligamental end. A pair of scissors indicates the transection site. ACL: anterior cruciate ligament; PCL: posterior cruciate ligament. (**B**) Male Gli1ER/Td mice received Tam injections (day −5 ~ −1) and meniscus injury (day 0) at 12 weeks of age. Knee joints were harvested at 1, 2, 4, 8, 12 weeks after injury. (**C**) Representative safranin O/fast green staining (top) and fluorescence images (bottom) of oblique sections of mouse

*Figure 4 continued on next page*

*Figure 4 continued*

knee joints harvested at indicated time points after injury. Dashed lines outline the meniscus. Scale bars, 200 μm. F: femur; T: tibia; S: synovium; MS: meniscus synovial end; ML: meniscus ligamental end. Red: Td; Blue: DAPI. (D) Repair score was evaluated at indicated time points after meniscus injury. n = 8 mice/group. (E) Cell density in the synovial and ligamental ends of meniscus was quantified at 4 weeks post meniscus injury. n = 5–6 mice/group. (F) The percentage of Td$^+$ cells in the synovial and ligamental ends of meniscus was also quantified. n = 5–6 mice/group. (G) qRT-PCR analysis of Hh signaling component genes in meniscus at 1 and 2 weeks post injury. n = 4 independent experiments. (H) Top panel is a schematic representation of the study protocol. Gli1ER/Td mice at 12 weeks of age were treated with Tam (day −5 to −1), meniscus injury (day 0), and EdU injections (day 9–13). A representative confocal image of knee joint at day 14 is shown below (Scale bars, 250 μm). Boxed areas of synovial and ligamental ends of meniscus (MS and ML, respectively) are shown at high magnification at the bottom (Scale bar, 25 μm). Green: EdU. (I) Top panel is a schematic representation of the study protocol. Gli1ER/Td (Ctrl) or Gli1ER/Td/DTA (DTA) mice were treated with Tam (day −5 to −1) and meniscus injury at 12 weeks of age (day 0). Representative fluorescent images of sagittal and coronal mouse knee joint sections at day 0 without injury are shown below (Scale bars, 200 μm). A: anterior; P: posterior. (J) The percentage of Td$^+$ cells in the anterior horn was quantified. n = 4 mice/group. (K) Representative safranin O/fast green staining (a, b) and fluorescence images (c, d) of oblique sections of mouse knee joints harvested at 12 weeks after injury. Dashed lines outline the meniscus. Scale bars, 200 μm. (L) Repair score was evaluated. n = 5 mice/group. Statistical analysis was performed using one-way ANOVA with Dunnett's post-hoc test for (D), one-way ANOVA with Tukey-Kramer post-hoc test for (G) and unpaired two-tailed t-test for (E), (F), (J) and (L). Data presented as mean ± s.e.m. *p<0.05, ***p<0.001.

The online version of this article includes the following source data and figure supplement(s) for figure 4:

**Source data 1.** Raw data for *Figure 4D,E,F,G,J,L*.
**Figure supplement 1.** Aging diminishes Gli1-lineage cell expansion and the repair ability of meniscus.
**Figure supplement 2.** Gli1$^+$ cells appear in proliferative cell clusters of degenerated human meniscus.

an expansion of Gli1$^+$ cells in human meniscus tissues and a potential action of Hh signaling in human meniscus repair.

## Activation of Hh/Gli1 pathway accelerates mouse meniscus repair

Since Gli1$^+$ cells and their descendants were greatly expanded at the early phase of meniscus injury repair, we hypothesized that activation of Hh/Gli1 pathway could stimulate the repair process. We adopted two approaches to test this hypothesis. One was to inject Gli1$^+$ cells freshly isolated from Gli1ER/Td meniscus into injured knees (*Figure 5A*). Strikingly, a single injection of cells right after injury resulted in a reconnection of the synovial and ligamental ends of injured meniscus at 4 weeks, leading to a repair score of 4.8 (*Figure 5B,C*). At the same time, these two meniscus ends were well separated in both vehicle and Gli1$^-$ meniscus cell-treated groups, with a repair score of only 1.8 and 1.9, respectively. Polarizing images clearly showed a disconnection of collagen fibers in mice that received either vehicle or Gli1$^-$ cells. However, in Gli1$^+$ cell-treated mice, collagen fibers crossed the broken ends of the meniscus, suggesting that the repair does occur at the structural level. Fluorescence imaging revealed that injected Gli1$^+$ cells expand and contribute to the newly formed connection at the injury site (*Figure 5D*).

In another approach, we injected PMA to the knee joint right after injury (*Figure 5E*). Four weeks later, the injured ends of PMA-treated meniscus were reconnected based on gross morphology, safranin O/fast green staining, and imaging of collagen fibers (*Figure 5F*), leading to a repair score of 4.9 (*Figure 5G*). There were more Td$^+$ meniscus cells in PMA-treated joints than vehicle-treated joints at 1 week after injury (*Figure 5H*). Hh signaling is known to induce osteogenesis (*Long et al., 2004*). MicroCT scanning of mouse joints detected no change in the calcified meniscus volume and a trend of decrease in osteophyte volume at 3 months post surgery and pur treatment (*Figure 5— figure supplement 1*), suggesting that Hh agonist does not affect overall joint calcification. These data clearly indicated a therapeutic effect of activating Hh signaling.

## Meniscus repair by enhancing Hh/Gli1 pathway delays OA progression

Meniscal injury inevitably leads to OA in human. To mimic this in mice, we characterized articular cartilage phenotype at 8 weeks post injury. Similar to the surgical destabilization of the medial meniscus (DMM) model of OA, our meniscus injury model caused cartilage degeneration, such as partially loss of proteoglycan, surface fibrillation, and reduction in uncalcified cartilage thickness at 2 months post injury (*Figure 6A,B*). Meanwhile, the calcified cartilage layer was not eroded (*Figure 6B*), suggesting a moderate OA with a Mankin Score of 6.9 (*Figure 6C*). Strikingly, injections of either Gli1$^+$ cells or PMA greatly reduced cartilage degeneration by retaining proteoglycan content, cartilage surface

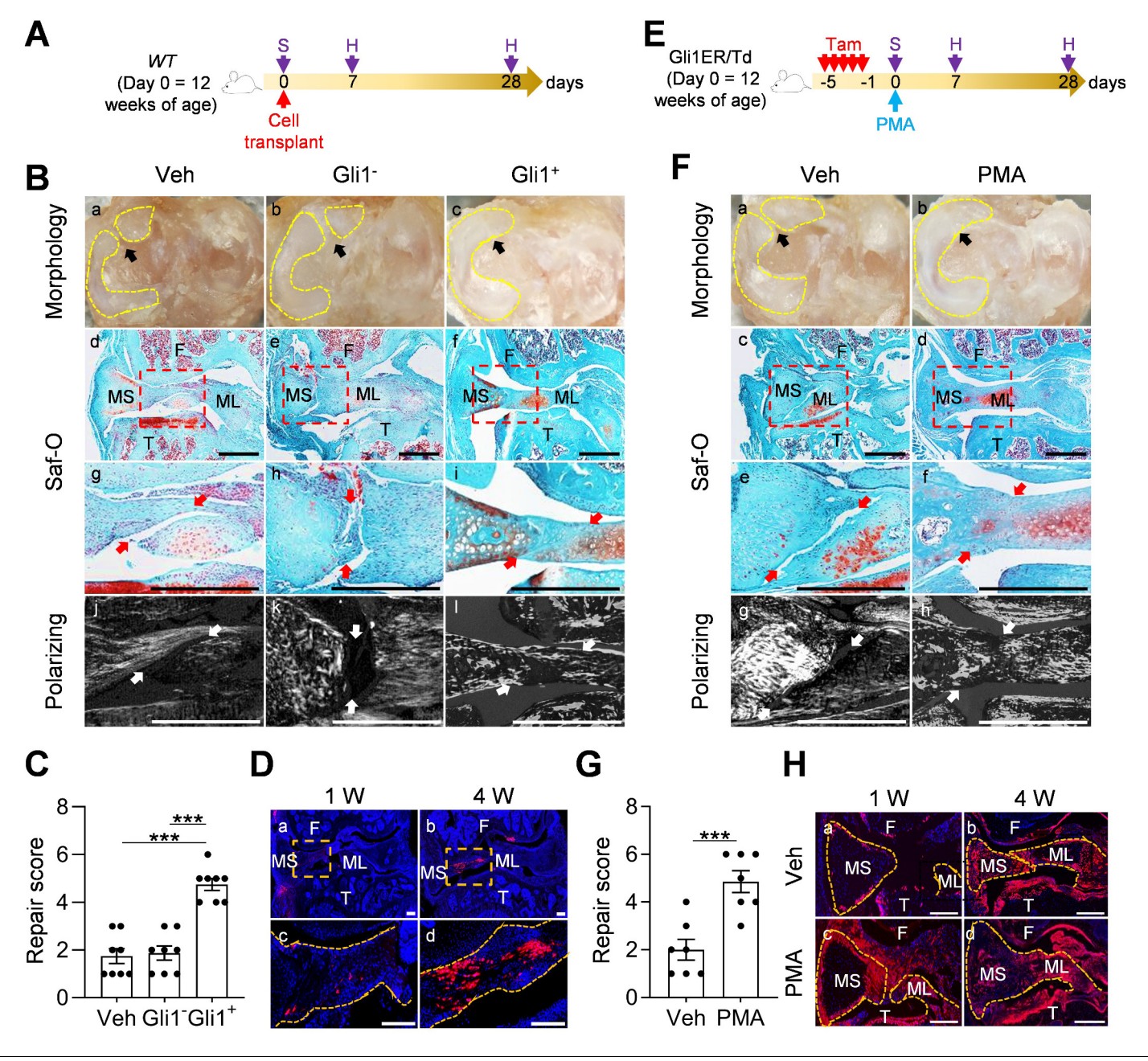

**Figure 5.** Activation of Hh/Gli1 pathway accelerates mouse meniscus repair. (A) Schematic representation of the study protocol. *WT* mice received meniscus injury at 12 weeks of age followed by transplantation of 5000 Gli1$^+$ or Gli1$^-$ meniscus cells at the injury site. Knee joints were harvested at 1 and 4 weeks after injury. (B) Representative overview (a–c), safranin O/fast green staining (d–i), and polarizing images (j–l) of mouse knee joints at 4 weeks after injury. Yellow dashed lines outline the overview morphology of injured meniscus. Meniscus is shown attached to tibial plateau. Arrows point to the injury site. Red boxed areas in d-f are shown at high magnification in g-i, respectively. Scale bars, 200 µm. F: femur; T: tibia; MS: meniscus synovial end; ML: meniscus ligamental end. (C) Repair score was evaluated. n = 8 mice/group. (D) Representative confocal images of mouse knee joints at 1 and 4 weeks after injury. Boxed areas in the top panel are shown at a high magnification at the bottom panel. Dashed line outlines meniscus. Scale bars, 200 µm. Blue: DAPI, Red: Td. (E) Schematic representation of the study protocol. Gli1ER/Td mice received Tam injections and meniscus injury at 12 weeks of age (day 0) followed by vehicle and PMA injection. Knee joints were harvested at 1 and 4 weeks after injury. (F) Representative overview (a, b), safranin O/fast green staining (c–f), and polarizing images (g, h) of mouse knee joints at 4 weeks after injury. Yellow dashed lines outline the overview morphology of injured meniscus. Meniscus is shown attached to tibial plateau. Red boxed areas in c and d are shown at high magnification in e and f, respectively. Arrows point to the injury site. Scale bars, 200 µm. (G) Repair score was evaluated. n = 7 mice/group. (H) Representative fluorescence images of vehicle- and PMA-treated mouse meniscus at 1 and 4 weeks after injury. Scale bars, 200 µm. Statistical analysis was performed using one-way ANOVA with Tukey-Kramer post-hoc test for (C) and unpaired two-tailed t-test for (G). Data presented as mean ± s.e.m. ***p<0.001.
*Figure 5 continued on next page*

*Figure 5 continued*

The online version of this article includes the following source data and figure supplement(s) for figure 5:

**Source data 1.** Raw data for *Figure 5C and G*.

**Figure supplement 1.** Hh agonist treatment does not promote joint calcification.

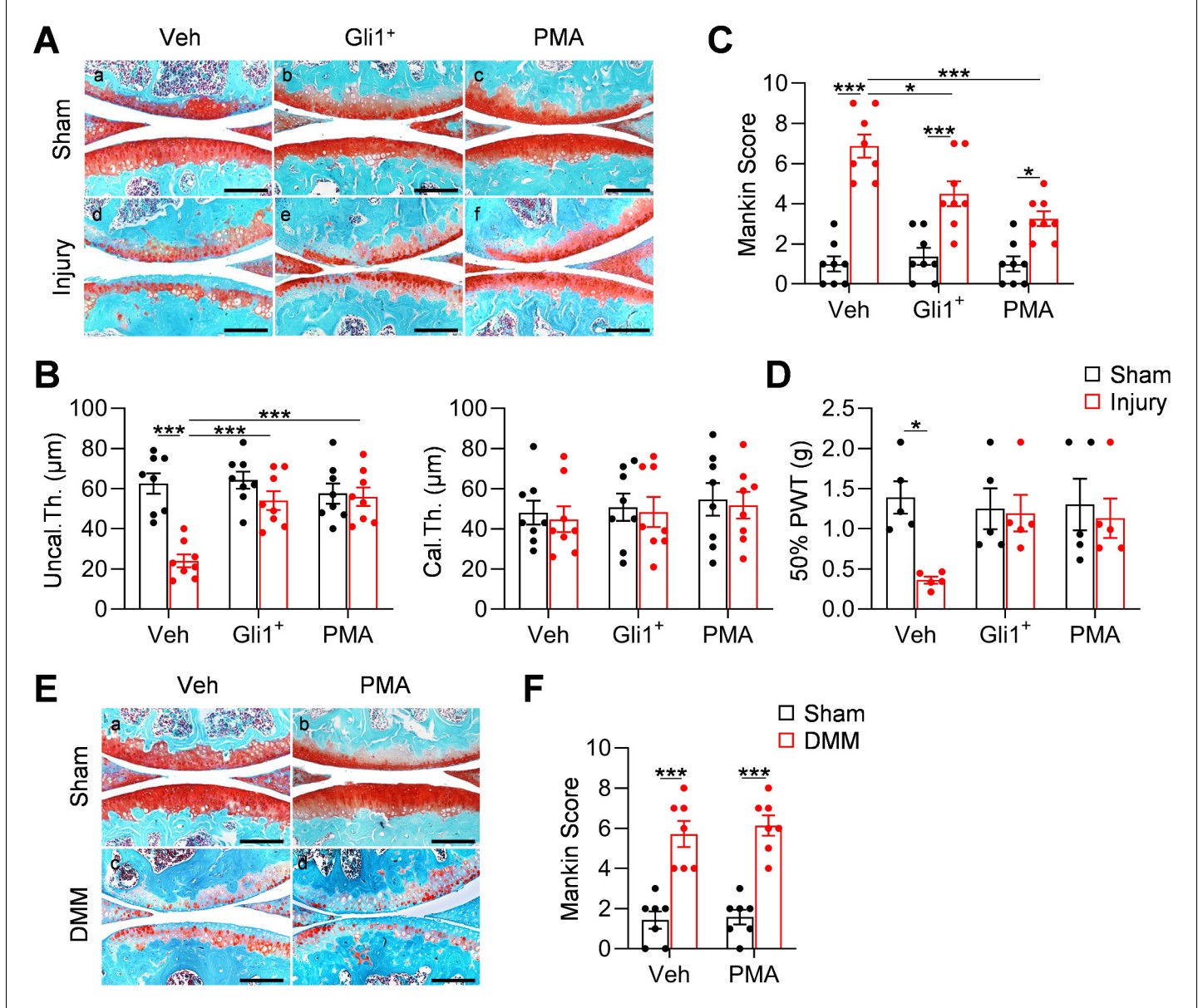

**Figure 6.** Meniscus repair by enhancing Hh/Gli1 signaling delays OA progression. (**A**) Representative safranin O/fast green staining of sagittal sections of vehicle-, Gli1+ cell- and PMA-treated mouse knee joints at 8 weeks after sham or meniscus injury. Scale bars, 200 μm. (**B**) Average thicknesses of uncalcified zone (Uncal.Th.) and calcified zone (Cal.Th.) of the tibial articular cartilage were quantified. n = 8 mice/group. (**C**) The OA severity was measured by Mankin score. n = 8 mice/group. (**D**) Von Frey assay was performed at 8 weeks after injury. PWT: paw withdrawal threshold. n = 5 mice/group. (**E**) Representative safranin O/fast green staining of sagittal sections of vehicle- and PMA-treated mouse knee joints at 8 weeks after sham or DMM surgery. Scale bars, 200 μm. (**F**) The OA severity was measured by Mankin score. n = 7 mice/group. Statistical analysis was performed using two-way ANOVA with Tukey-Kramer post-hoc test. Data presented as mean ± s.e.m. *p<0.05, **p<0.01, ***p<0.001.

The online version of this article includes the following source data for figure 6:

**Source data 1.** Raw data for *Figure 6B,C,D and F*.

smoothness, and the structure of uncalcified cartilage. These treatments led to a reduction in Mankin Score by 35% and 53%, respectively. Von Frey assay is commonly used in OA study as a pain outcome by evaluating mechanical allodynia. Using this assay, we observed that OA knees show reduced paw withdraw threshold compared to sham knees. However, this OA-related pain was mostly attenuated in Gli1$^+$ cell- or PMA-treated knees (*Figure 6D*).

Hh signaling has also been indicated to play a role in the development of articular cartilage (*Kurio et al., 2018*) and in OA progression (*Lin et al., 2009*). To exclude the possibility that activating Hh signaling directly affects OA progress, we performed DMM surgery in 3-month-old male *WT* mice and injected PMA into their knee joints right after surgery. Two months later, we observed a similar level of cartilage degeneration in control and treated mice (*Figure 6E,F*), suggesting that the effect of Hh signaling on OA development is mediated through meniscus repair but not through directly acting on cartilage. It is worthwhile noting that different from our transient activation approaches, the previous conclusion about the catabolic action of Hh signaling on cartilage is derived from constant modulation of this signaling by genetic approaches (*Lin et al., 2009*).

## Discussion

Previous studies have identified the existence of mesenchymal progenitors in meniscus based on their clonogenic and multilineage differentiation activities in culture. However, their in vivo properties and regulatory signals are largely unknown. In this work, by using a lineage tracing mouse line, cell culture, and a meniscus injury model, we demonstrated that Gli1$^+$ cells are a major subset of meniscal mesenchymal progenitors and that those Hh-responsive cells are critical for meniscus development and injury response. Moreover, aging reduces this Gli1$^+$ progenitor population in healthy meniscus as well as their expansion after injury, which is consistent with attenuated healing in old mice. On the therapeutic side, the activation of Hh/Gli1 signaling in adult meniscus leads to accelerated meniscus healing process and the delay of OA changes, indicating a protective role of Hh signaling on meniscus against degeneration.

Hh signaling plays a key role during embryonic development and tissue patterning. Meniscus is somewhat similar to periodontal ligament (PDL) and enthesis in skeleton. Prior studies have shown that Gli1$^+$ PDL cells contribute to the formation of periodontal tissues and can regenerate alveolar bone (*Hosoya et al., 2020*). Many studies also demonstrated the importance of Gli1 lineage cells as the progenitors in the development and regeneration of enthesis (*Dyment et al., 2015*; *Schwartz et al., 2017*; *Felsenthal et al., 2018*). In long bones, embryonic Gli1$^+$ cells give rise to multiple cell types associated with the skeleton and are a major source of osteoblasts in both fetal and postnatal life of the mouse (*Shi et al., 2017*). We also discovered that Gli1$^+$ mesenchymal progenitors from neonatal periarticular surfaces are capable of generating mesenchymal lineage cells, including osteoblasts, osteocytes, and adipocytes in the secondary ossification center of long bones (*Tong et al., 2019*). Surprisingly, Gli1$^+$ cells do not contribute to the early development of meniscus. They start to appear from 2 weeks of age first in the anterior horn and later in the posterior horn. The different Gli1-labeling patterns in the two horns may reflect the distinct cellular composition of anterior and posterior horns reported previously (*Gamer et al., 2017b*). Since menisci undergo rapid growth postnatally and Gli1$^+$ cells are absent from the meniscus body, our data indicated that there must be other distinct progenitor population(s) contributing to meniscus development. Indeed, we found that Gli1$^-$ meniscus cells are also able to proliferate and differentiate in vitro albeit with less activities compared to Gli1$^+$ cells.

At the adult stage, we found that Gli1$^+$ cells are mainly located at the superficial layer of meniscal horns. They rarely contribute to the inner cells of meniscus probably due to the low turnover of meniscus tissue. However, similar to their counterparts in the periosteum (*Shi et al., 2017*) and tendon enthesis (*Schwartz et al., 2015*), they play an important role in tissue regeneration. In our study, we established a meniscus injury model by transection of the anterior horn. A previously reported mouse meniscus injury model (meniscectomy of the anterior horn) revealed almost complete regeneration of meniscus and only subtle cartilage degeneration at 6 weeks post surgery (*Hiyama et al., 2017*). Compared to that, meniscus repair in our injury model is slow and inefficient with disconnected collagen fibers remaining at the injured site at 3 months post surgery. This prolonged injury causes damage to articular cartilage, leading to moderate OA. Hence, our model is suitable to study

the beneficial effects of Hh signaling on meniscus repair and meniscus damage-related OA progression.

Notably, we also observed a quick expansion of synovium enriched with Gli1-lineage cells at the early stage of repair. Therefore, we cannot exclude the possibility that synovial Gli1$^+$ cells also contribute to meniscus regeneration. That being said, our data showed that Gli1$^+$ primary meniscus cells injected into knee joints incorporate into meniscus tissue and accelerate repair, indicating that the endogenous Gli1$^+$ meniscus cells are likely responsible for the repair. Our studies found that Ihh is the most up-regulated Hh ligand during development and that Dhh is the one upregulated after injury. We have not investigated the cell source of Hh ligand in meniscus. Since Ihh from the prehypertrophic chondrocytes is known for regulating long bone development through endochondral ossification (*Olsen et al., 2000*), it is possible that fibrochondrocytes in the deep layer of meniscus produce the Hh signals.

Our lineage tracing and injury studies are based on mouse models. However, rodents are different from human by having bony ossicles in the meniscus horns (*Shaw and Martin, 1962*). To show the clinical relevance of our research, we first demonstrated that porcine meniscus has similar anatomic distribution of Gli1$^+$ cells, suggesting a conservation of this patterning between species. While we did not detect Gli1$^+$ cells in healthy human meniscus, likely due to the sample being collected from the body rather than the horns, we found similar expansion of Gli1$^+$ cells in cell clusters of diseased meniscus, suggesting the translatability of our findings. Our studies, therefore, have uncovered a critical role of Hh/Gli1 signaling in knee meniscus development and regeneration and provide evidence for targeting this pathway as a novel meniscus injury therapy and potentially for preventing OA development.

# Materials and methods

## Key resources table

| Reagent type (species) or resource | Designation | Source or reference | Identifiers | Additional information |
|---|---|---|---|---|
| Strain, Strain background (*Mus musculus*) | Gli1$^{CreER}$ | Jackson Laboratory | Stock No: 007913 | |
| Strain, Strain background (*Mus musculus*) | Rosa26 $^{lsl-tdTomato}$ | Jackson Laboratory | Stock No: 007909 | |
| Strain, Strain background (*Mus musculus*) | Rosa26 $^{lsl-DTA}$ | Jackson Laboratory | Stock No: 010527 | |
| Sequence-based reagent | Gli1$^{CreER}$ Primer 1 | Jackson Laboratory | PCR Primer | 5'-GCGGTCTGGC AGTAAAAACTATC-3' |
| Sequence-based reagent | Gli1$^{CreER}$ Primer 2 | Jackson Laboratory | PCR Primer | 5'-GTGAAACAGCAT TGCTGTCACTT-3' |
| Sequence-based reagent | Gli1$^{CreER}$ Primer 3 | Jackson Laboratory | PCR Primer | 5'-CACGTGGGCT CCAGCATT-3' |
| Sequence-based reagent | Gli1$^{CreER}$ Primer 4 | Jackson Laboratory | PCR Primer | 5'-TCACCAGTCAT TTCTGCCTTTG-3' |
| Sequence-based reagent | Rosa26 $^{lsl-tdTomato}$ Primer 1 | Jackson Laboratory | PCR Primer | 5'-AAGGGAGC TGCAGTGGAGTA-3' |
| Sequence-based reagent | Rosa26 $^{lsl-tdTomato}$ Primer 2 | Jackson Laboratory | PCR Primer | 5'-CCGAAAATCTGT GGGAAGTC-3' |
| Sequence-based reagent | Rosa26 $^{lsl-tdTomato}$ Primer 3 | Jackson Laboratory | PCR Primer | 5'-GGCATTAAA GCAGCGTATCC-3' |
| Sequence-based reagent | Rosa26 $^{lsl-tdTomato}$ Primer 4 | Jackson Laboratory | PCR Primer | 5'-CTGTTCCTGT ACGGCATGG-3' |
| Sequence-based reagent | Rosa26 $^{lsl-DTA}$ Primer1 | Jackson Laboratory | PCR Primer | 5'-CGACCTGCA GGTCCTCG-3' |
| Sequence-based reagent | Rosa26 $^{lsl-DTA}$ Primer 2 | Jackson Laboratory | PCR Primer | 5'-CCAAAGTCGCT CTGAGTTGTTATC-3' |
| Sequence-based reagent | Rosa26 $^{lsl-DTA}$ Primer 3 | Jackson Laboratory | PCR Primer | 5'-GAGCGGGA GAAATGGATATG-3' |
| Sequence-based reagent | Rosa26 $^{lsl-DTA}$ Primer 4 | Jackson Laboratory | PCR Primer | 5'-CTCGAGTTTG TCCAATTATGTCAC-3' |

*Continued on next page*

*Continued*

| Reagent type (species) or resource | Designation | Source or reference | Identifiers | Additional information |
|---|---|---|---|---|
| Antibody | Rabbit polyclonal Anti-Gli1 | NOVUS biologicals | NB600-600 | IF (1:100) |
| Antibody | Rabbit polyclonal Anti-Ki67 | Abcam | ab15580 | IF (1:50) |
| Antibody | Rat monoclonal Anti-Sca1 | Santa Cruz Bio | sc-52601 | IF (1:200) |
| Antibody | Rat monoclonal Anti-Cd200 | Santa Cruz Bio | sc-53100 | IF (1:100) |
| Antibody | Mouse monoclonal Anti-Cd90 | Santa Cruz Bio | sc-53456 | IF (1:100) |
| Antibody | Mouse monoclonal Anti-PDGFRα | Santa Cruz Bio | sc-398206 | IF (1:200) |
| Antibody | Mouse monoclonal Anti-Cd248 | Santa Cruz Bio | sc-377221 | IF (1:200) |
| Antibody | Rabbit polyclonal anti-Prg4 | Abcam | ab28484 | IF (1:50) |
| Antibody | Rat monoclonal Anti-Sca1 | BioLegend | 108131 | Flow analysis (1:100) |
| Antibody | Mouse monoclonal Anti-Cd90 | BioLegend | 202526 | Flow analysis (1:100) |
| Antibody | Rat monoclonal Anti-Cd200 | BioLegend | 123809 | Flow analysis (1:100) |
| Antibody | Rat monoclonal Anti-PDGFRα | BioLegend | 135907 | Flow analysis (1:100) |
| Software, algorithm | Graphpad 8.0 | | Statistical Analysis | Graph preparation, statistical analysis |

## Animals

All animal work performed in this report was approved by the Institutional Animal Care and Use Committee (IACUC) at the University of Pennsylvania. $Gli1^{CreER}$ $Rosa26^{lsl-tdTomato}$ (Gli1ER/Td) mice were generated by breeding $Gli1^{CreER}$ mice (Jackson Laboratory, Bar Harbor, ME USA; 007913) with $Rosa26^{lsl-tdTomato}$ mice (Jackson Laboratory; 007909). They were further bred with $Rosa26^{lsl-DTA}$ mice (Jackson Laboratory; 010527) to produce $Gli1^{CreER}$ $Rosa26^{lsl-tdTomato}$ $Rosa26^{lsl-DTA}$ (Gli1ER/Td/DTA). In accordance with the standards for animal housing, mice were group housed at 23–25°C with a 12 hr light/dark cycle and allowed free access to water and standard laboratory pellets. All animal work performed in this report were approved by the Institutional Animal Care and Use Committee (IACUC) at the University of Pennsylvania.

To induce Td expression or ablate Gli1$^+$ cells, mice (Gli1ER/Td or Gli1ER/Td/DTA) received vehicle or Tamoxifen (Tam) injections at 50 mg/kg at P4 and P5 or 75 mg/kg for 5 days at ages older than 1 week. For EdU labeling of proliferating cells, mice were injected with daily 1.6 mg/kg EdU (Invitrogen, Carlsbad, USA, A10044) for 4 days before harvesting. For EdU labeling of slow-cycling cells, mice were injected with daily 5 mg/kg EdU for 4 days at P3-6.

Male mice at 3 months of age were subjected to meniscus injury at right knees. To perform the surgery, the joint capsule was opened immediately after anesthesia and the anteriomedial horn of meniscus was cut into two parts using microsurgical scissors. The joint capsule and the subcutaneous layer were then closed with suture followed by skin closure with tissue adhesive. In sham surgery, meniscus was visualized but not transected. For cell treatment, cells digested from meniscus of Gli1ER/Td mice were sorted by FACS to collect Td$^+$ and Td$^-$ cells. A total of 5000 cells were injected into the knee joint space of sibling *WT* mice immediately after meniscus surgery. For activator treatment, 2 µL purmorphamine (PMA, 100 µM) were injected into the knee joint space of *WT* or Gli1ER/Td mice immediately after surgery. Mice were euthanized at indicated time points for histology analysis.

The knee joint pain after meniscus injury was evaluated in mice at 1 month after surgery using von Frey filaments as described previously (*Jia et al., 2016*). An individual mouse was placed on a wire-mesh platform (Excellent Technology Co.) under a 4 × 3 × 7 cm cage to restrict their move. Mice were trained to be accustomed to this condition every day starting from 7 days before the test. During the test, a set of von Frey fibers (Stoelting Touch Test Sensory Evaluator Kit #2 to #9; ranging from 0.015 to 1.3 g force) were applied to the plantar surface of the hind paw until the fibers bowed, and then held for 3 s. The threshold force required to elicit withdrawal of the paw (median 50% withdrawal) was determined five times on each hind paw with sequential measurements separated by at least 5 min.

To surgically induce OA, male mice at 3 months of age were subjected to the surgical destabilization of the medial meniscus (DMM) surgery at right knees and sham surgery at left knees as described previously (*Zhang et al., 2014*). Briefly, in DMM surgery, the joint capsule was opened immediately after anesthesia and the medial meniscotibial ligament was cut to destabilize the meniscus without damaging other tissues. In sham surgery, the joint capsule was opened in the same fashion but without any further damage.

## Human and Mini-pig meniscus samples

The meniscus samples were prepared from the de-identified specimens obtained at the total arthroplasty of the knee joints and used for histological and immunohistochemical examination. The meniscus degeneration severity was evaluated according to the meniscus surface including lamellar layer, cellularity, collagen organization and safranin O/fast green staining (*Pauli et al., 2011*). 6-month-old male Yucatan minipigs were utilized (Sinclair Bioresources) to provide meniscus tissues. Anterior horn meniscus tissue was obtained for following histological analysis.

## Histology

After euthanasia, mouse knee joints were harvested and fixed in 4% PFA overnight followed by decalcification in 10% M EDTA (pH 7.4). Samples were processed for either cryosections after 1 week of decalcification or paraffin sections after 4 weeks of decalcification. For healthy knee joints, a series of 6-µm-thick sections were cut across the entire compartment of the joint at the coronal or sagittal plane followed by fluorescent imaging (cryosections) or safranin O/fast green staining for brightfield imaging (paraffin sections). For meniscus injured knee joints, a series of 6-µm-thick sections were cut across the entire anterior horn area in the direction perpendicular to the meniscus injury gap (oblique sections) followed by fluorescent imaging (cryosections) or safranin O/fast green staining for brightfield imaging (paraffin sections). To evaluate meniscus healing process, we collected all sections (~15) including both synovial and ligamental ends. Three sections were selected from each knee, corresponding to 1/3 (sections 1–5), 2/3 (sections 6–10), and 3/3 (sections 11–15) regions of the entire section set to quantify the meniscus repair scores according to the connection between two ends, existence of fibrochondrocyte and sensitivity of safranin O staining (*Ishida et al., 2007*). The method to measure Mankin Score was described previously (*Aigner et al., 2010*). Briefly, two sections within every consecutive six sections in the entire sagittal section set for each knee were stained with safranin O/fast green and scored by two blinded observers (YW and HS). Each knee received a single score representing the maximal score of its sections.

For immunohistochemistry staining, mouse, porcine, and human paraffin sections were incubated with rabbit anti-Gli1 (NOVUS biologicals, NB600-600) and anti-Ki67 (Abcam, ab15580) at 4°C overnight followed by binding with biotinylated secondary antibody incubation for 1 hr and DAB color development. For immunofluorescence staining, sagittal mouse knee joint cryosections were incubated with rat anti-Sca1 (Santa cruz, sc-52601), rat anti-Cd200 (Santa cruz, sc-53100), mouse anti-Cd90 (Santa cruz, sc-53456), mouse anti-PDGFRα (Santa cruz, sc-398206), mouse anti-Cd248 (Santa cruz, sc-377221), rabbit anti-Prg4 (Abcam, ab28484) at 4°C overnight followed by binding with corresponding Alexa Fluor 488-conjugated secondary antibody incubation for 2 hr and DAPI counterstaining.

## MicroCT analysis

The distal femur and proximal tibia of mouse knee joints was scanned at a 6 µm isotropic voxel size with a microCT 35 scanner (Scanco Medical AG, Brüttisellen, Switzerland). All images were smoothened by a Gaussian filter (sigma = 1.2, support = 2.0). Sagittal images were contoured to measure calcified meniscus volume, while coronal images were contoured to measure osteophyte volume surrounding distal femur and proximal tibia.

## Primary mouse meniscus cell culture

The entire mouse menisci were dissected from 4-week-old mice and digested in 0.25% Trypsin-EDTA (Gibco) for 1 hr followed by 300 U/mL collagenase type I (Worthington Biochemical) for 2 hr. Cells from the second digestion were cultured in the growth medium (αMEM supplemented with 10% fetal bovine serum plus 100 IU/mL penicillin and 100 mg/mL streptomycin) to obtain meniscus

cell culture. For CFU-F assay, digested cells were seeded at 20,000 cells per T25 flask. Seven days later, flasks were stained with 3% crystal violet to quantify colony numbers. To study cell migration, primary meniscus cells were seeded in 12-well plates. When reaching confluency, the cell layer was scratched by a 1000 µL pipette tip and then cultured in FBS-free growth medium. Wound closure was monitored by imaging at 0 and 48 hr later. To study cell proliferation, primary meniscus cells were seeded at 50,000 cells/well in 12-well plates and cell numbers were counted 2, 4, and 6 days later. Chondrogenic, osteogenic, and adipogenic differentiation was performed as describe previously (*Wang et al., 2019*). The working concentration of GANT-61 and PMA was 10 and 1 µM, respectively.

## Flow cytometry and cell sorting

Flow cytometry and cell sorting were performed on a FACS Aria III cell sorter (BD Biosciences) and analyzed using FlowJo software (Tree Star). Digested meniscus cells were re-suspended in flow buffer (2% FBS/PBS) and stained with Sca1 (BioLegend, 108131), Cd90 (BioLegend, 202526), Cd200 (BioLegend, 123809), and PDGFRα (BioLegend, 135907) flow antibody for 1 hr at 4°C. After PBS wash, cells were analyzed by flow cytometry or sorted for $Gli1^+$ and $Gli1^-$ cells.

## RNA analysis

To quantify the expression level of marker genes, total RNA was collected in Tri Reagent (Sigma, St. Louis, MO, USA) for RNA purification. A Taqman Reverse Transcription Kit (Applied BioSystems, Inc, Foster City, CA, USA) was used to reverse transcribe mRNA into cDNA. The power SYBR Green PCR Master Mix Kit (Applied BioSystems, Inc) was used for quantitative real-time PCR (qRT-PCR). The primer sequences for the genes used in this study are listed in *Supplementary file 1*.

## Statistical analyses

Data are expressed as means ± standard error of the mean (SEM) and analyzed by t-tests, one-way ANOVA with Dunnett's or Tukey-Kramer posttest and two-way ANOVA with Tukey-Kramer post-test for multiple comparisons using Prism eight software (GraphPad Software, San Diego, CA). For assays using primary cells, experiments were repeated independently at least three times and representative data were shown here. Values of $p < 0.05$ were considered statistically significant.

## Acknowledgements

This study was supported by NIH grants NIH/NIA R01AG067698 (to L.Q.), P30AR069619 (to Penn Center for Musculoskeletal Disorders), and NSF CMMI-1751898 (to LH).

## Additional information

### Funding

| Funder | Grant reference number | Author |
| --- | --- | --- |
| National Institutes of Health | R01AG067698 | Ling Qin |
| National Institutes of Health | P30AR069619 | Ling Qin |
| National Science Foundation | CMMI-1751898 | Lin Han |

The funders had no role in study design, data collection and interpretation, or the decision to submit the work for publication.

### Author contributions

Yulong Wei, Hao Sun, Tao Gui, Conceptualization, Data curation, Formal analysis, Validation, Investigation, Visualization, Methodology, Writing - original draft, Project administration; Lutian Yao, Leilei Zhong, Wei Yu, Conceptualization, Data curation, Formal analysis, Methodology, Project administration; Su-Jin Heo, Conceptualization, Data curation, Formal analysis, Methodology; Lin Han, Conceptualization, Supervision, Funding acquisition, Investigation, Methodology, Writing - review and editing; Nathaniel A Dyment, Conceptualization, Methodology, Writing - review and editing;

Xiaowei Sherry Liu, Yejia Zhang, Eiki Koyama, Fanxin Long, Conceptualization, Supervision, Investigation, Methodology, Writing - review and editing; Miltiadis H Zgonis, Conceptualization, Resources, Supervision, Investigation, Methodology; Robert L Mauck, Conceptualization, Resources, Supervision, Investigation, Methodology, Writing - review and editing; Jaimo Ahn, Conceptualization, Supervision, Investigation, Methodology, Writing - original draft, Writing - review and editing; Ling Qin, Conceptualization, Resources, Data curation, Supervision, Funding acquisition, Investigation, Methodology, Writing - original draft, Writing - review and editing

## Author ORCIDs

Yulong Wei (iD) https://orcid.org/0000-0003-3823-9984
Leilei Zhong (iD) http://orcid.org/0000-0003-1153-4115
Wei Yu (iD) http://orcid.org/0000-0001-6705-8264
Fanxin Long (iD) http://orcid.org/0000-0001-9785-5379
Ling Qin (iD) https://orcid.org/0000-0002-2582-0078

## Ethics

Animal experimentation: All animal work performed in this report was approved by the Institutional Animal Care and Use Committee, All of the animals were handled according to approved institutional animal care and use committee (IACUC) protocols (#803233) of the University of Pennsylvania. The protocol was approved by the Penn Animal Welfare/institutional animal care and use committee of the university of Pennsylvania.

## Decision letter and Author response

Decision letter https://doi.org/10.7554/eLife.62917.sa1
Author response https://doi.org/10.7554/eLife.62917.sa2

# Additional files

## Supplementary files

• Supplementary file 1. Mouse real-time PCR primer sequences.

• Transparent reporting form

## Data availability

All data generated or analysed during this study are included in the manuscript and supplementary files. Source data files have been provided for Figures 1 to 6.

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
