## [Decision Letter]

**Acceptance summary:**

The authors provide evidence that Hedgehog activation promotes meniscal healing and identify population of cells that are positive for Gli1, an effector protein in the Hedgehog signaling pathway. Using a combination of approaches, including lineage tracing, in vitro cell culture approaches and cell transplantation experiments, they show that this Gli1 populations contains putative meniscal progenitors involved in meniscus development and healing.

**Decision letter after peer review:**

Thank you for submitting your article "Identification of Gli1 as a progenitor cell marker for meniscus development and injury repair" for consideration by *eLife*. Your article has been reviewed by 3 peer reviewers, and the evaluation has been overseen by a Reviewing Editor and Kathryn Cheah as the Senior Editor. The following individual involved in review of your submission has agreed to reveal their identity: Stavros Thomopoulos (Reviewer #2).

The reviewers have discussed the reviews with one another and the Reviewing Editor has drafted this decision to help you prepare a revised submission.

Summary:

Injuries of the meniscus are associated with the future development of articular cartilage damage and ultimately osteoarthritis (OA). Prior work in this field has suggest that there are undifferentiated progenitor cells residing in the meniscus and it has been hypothesized that these cells could be harnessed to aid in meniscal healing after injury. The authors provide evidence that Hedgehog activation promotes meniscal healing and identify population of cells that are positive for Gli1, an effector protein in the Hedgehog signaling pathway. Using a combination of approaches, including lineage tracing, in vitro cell culture approaches and cell transplantation experiments, they show that this Gli1 populations contains putative meniscal progenitors involved in meniscus development and healing. Overall, the reviewers found this paper to have high potential and were particularly enthusiastic about the therapeutic potential of Purmorphamine to promote meniscal healing. However, all reviewers felt that the conclusions (and title) of this paper overstated the utility of Gli1 as a marker of de facto meniscal progenitor cells. It is specifically requested that the title of this manuscript be reworded in light of the previous work showing that Gli1 can be found in a number of cell types. In several places, additional work was requested to support the conclusions made in this manuscript. Please see the detailed comments below.

Essential revisions:

1. The authors describe many of their results as "novel". Gli1 reporter mice have been used extensively in other tissues to non-specifically describe progenitor cells (bone marrow, periosteum, peri-vascular spaces and others). Further, the role of Gli1+ cells in enthesis and periodontal ligament (PDL) formation and healing has been previously explored. Gli proteins, which have a half-life of minutes-to-hours, may be a relatively unstable foundation for defining cellular identity. While the value of Gli1 as a general Hh reporter is clear, its utility as a putative stem cell marker (Title) does not seem adequately substantiated. The authors must temper their statements on novelty, exclusivity and utility of Gli1. The title of this paper also should be reworded.

2. The Hedgehog (Hh) signaling manipulation conducted is rather straightforward and some overlapping studies have been performed in murine joints. Many of the experimental results could have been predicted. Other elements that contribute to the superficial nature of the studies are that Gli1 reporter activity is the only marker of Hh signaling examined (for example Gli2/Gli3 are not), and that the abundance and cellular source of an Hh ligand during development or repair is never entertained. Of note, these reporters for Ihh and Shh are available.

3. It is a stretch to say that Gli1;tdTom labels meniscus progenitor cells (Lines 268-271). There is relative enrichment of Sca1/CD90/CD200/PDGFRa in Gli1+ cells (Figure 2B), yet the vast majority of cells positive for those markers are Gli1-negative (Figure S5). Positive outcomes during in vitro differentiation and scratch assays may primarily result from increased Hh-mediated proliferation. This logic extends all the way through the in vivo experiments (which are quite promising, translationally).

4. The spatial profile of Gli1-expressing cells in the meniscus is beautifully described, however an interpretation for the superficially restricted zonation of Gli1 reporter activity is not given. Do these superficial cells have more or less cartilage antigen expression? Is there something clearly physiologically different in the Gli1-rich superficial layers that could be determined? Line 401 cites an osteoblast paper to set up the relevance of Gli1+ cells in development of musculoskeletal tissues. However, the meniscus is much more similar to the enthesis and the PDL. The authors should therefore lead with that literature. The PDL literature in particular is not cited and should be added. Also missing are recent enthesis development/regeneration papers (PMID: 30504126, 26141957, and 28219952).

5. The characterization of Gli1+ and Gli1- FAC sorted cells could be expanded on a bit.

6. CFU-F images should be provide in addition to quantification. The differentiation studies in Figure 2E are non-quantitative and not convincing. Further, it is a little contradictory that under certain contexts Gli1+ cells form more cartilage (2E), but under other culture conditions they have reduced cartilage markers (2F). These points need to be clarified.

7. In Figure 5, changes in distribution or survival of Gli1+/- cells may underlie the difference, but survival nor Gli1- cell distribution were not assessed.

8. Cartilage differentiation within the meniscus appears to be promoted with Gli1+ cell therapy and Purmorphamine. This could be assessed. Similarly, Hh signaling is known to induce osteogenesis. Osteoblastic antigens and/or presence of osteophytes should be assessed for in purmorphamine treated joints.

9. One topic that is not covered in the paper is the role of Hh signaling in chondrocyte mineralization. This has been well studied in the growth plate (esp. related to PTHrP / IHH feedback loop) and may have relevance to the meniscus as well. The healing studies should consider this carefully, as ectopic mineralization is a possible negative side effect of Hh treatment.

10. There are a number of places in the results where it is unclear if the authors are talking about Gli+ cells or Gli1-lineage cells. This should be clarified throughout, perhaps with specific nomenclature that defines "Gli1+" as cells that are positive for Gli and "Gli1-lineage" for cells that are descendants of Gli+ cells. Supplemental Figure 1A should be in the main document. Similar schematics in other figures are very useful for understanding the experiment.

11. What are the temporal expression patterns of Gli1 and other Hh related genes during development and healing? It would be informative to see localized expression (e.g., in situ hybridization) or qPCR expression for healing tissues.

12. The authors should clarify a number of things with meniscal cell isolation: (a) There are clearly differences in cell phenotype between superficial and deep areas and between attachment and midsection; was this considered for cell isolation? (b) I assume TAM injections were performed and then cells were isolated a few days later via FACS; please clarify details to show that Gli1+ (not Gli1-lineage) cells were characterized. (c) Figure 2: 3-month old mice were used, but again, Gli+ vs. Gli1-lineage cells is not indicated.

13. The mechanisms by which Gli1+ and Hh treatments work is not explored. Some of the results are counter-intuitive. For example, why would Hh stimulate proliferation if Gli1+ cells if these are thought to be slow turnover resident stem cells? Furthermore, why would Hh stimulation lead to proliferation rather than differentiation, (in contrast to what is know in growth plate biology)?

14. The assessment of healing is qualitative/semi-quantitative (histomorphometry). The authors should perform a more rigorous assessment of healing to demonstrate the effectiveness of the Gli1+ cell and Hh therapies. This should include quantitative outcome(s) such as qPCR, mechanics, etc.

15. The Gli1+ cell therapy histologic results are impressive. This is surprising because the delivery method was relatively simple. How much cell engraftment was there? Can the authors comment further (or add experiments to elucidate) on how long the cells were present and what their direct involvement was in healing?

16. The authors show that native Gli1+ cells expand after injury. If this is the case, what is the rationale for adding more Gli1+ cells? Is the idea that the tissue has the capacity to heal but there aren't enough native Gli1+ cells to do the job?

17. Figures and text jump between methodologies, making interpretation of results difficult. Figure 1 shows that superficial cells of the meniscus generally have active Hh signaling 24-hours prior to a variety of postnatal-to-adult timepoints (A, B, E, F), and postnatal Hh signaling drives proliferation of early meniscus cells (C, D). It does not appear to report any long-term pulse/chase lineage tracing experiment as suggested in the text (Lines 223+). If this interpretation is incorrect, perhaps this could be addressed by increased clarity of figures and text (Methods, Results, Figure organization and captions)?

---

## [Author Response]

Essential revisions:1. The authors describe many of their results as "novel". Gli1 reporter mice have been used extensively in other tissues to non-specifically describe progenitor cells (bone marrow, periosteum, peri-vascular spaces and others). Further, the role of Gli1+ cells in enthesis and periodontal ligament (PDL) formation and healing has been previously explored. Gli proteins, which have a half-life of minutes-to-hours, may be a relatively unstable foundation for defining cellular identity. While the value of Gli1 as a general Hh reporter is clear, its utility as a putative stem cell marker (Title) does not seem adequately substantiated. The authors must temper their statements on novelty, exclusivity and utility of Gli1. The title of this paper also should be reworded.

We thank the Reviewers for this suggestion. The new title is “The critical role of Hedgehog-responsive mesenchymal progenitors in meniscus development and injury repair”. We also revised our manuscript to accurately reflect the novelty, exclusivity, and utility of Gli1.

2. The Hedgehog (Hh) signaling manipulation conducted is rather straightforward and some overlapping studies have been performed in murine joints. Many of the experimental results could have been predicted. Other elements that contribute to the superficial nature of the studies are that Gli1 reporter activity is the only marker of Hh signaling examined (for example Gli2/Gli3 are not), and that the abundance and cellular source of an Hh ligand during development or repair is never entertained. Of note, these reporters for Ihh and Shh are available.

To address this comment, we analyzed the expression of Hh signaling components in mouse meniscus. qRT-PCR analyses of meniscus from mice at 1, 4, and 8 weeks of age and mice at 1 and 2 weeks post meniscal injury demonstrated that Hh signaling components are increased in adolescent mice and during early injury repair. Interesting, Hh ligands play distinct roles in the development and injury repair. While Ihh was the most up-regulated ligand during development, Dhh was the one up-regulated after injury. We did not detect Shh mRNA in our assays. Among Gli transcription factors, Gli1 was the most up-regulated one during development and after injury. Gli2 shared the similar expression pattern but its fold changes were much less than Gli1. We did not detect an increase of Gli3 expression at these time points. These data are now included as Figure 1E and 4F. In the future, we will acquire *Shh-CreGFP, Ptch1-lacZ, Gli1-lacZ*, and *Gli2-lacZ* mice from Jax Laboratory for further study. Unfortunately, Dr. Benjamin Humphreys replied that he discontinued his *Ihh-LacZ* mouse line ^(1)^.

3. It is a stretch to say that Gli1;tdTom labels meniscus progenitor cells (Lines 268-271). There is relative enrichment of Sca1/CD90/CD200/PDGFRa in Gli1+ cells (Figure 2B), yet the vast majority of cells positive for those markers are Gli1-negative (Figure S5). Positive outcomes during in vitro differentiation and scratch assays may primarily result from increased Hh-mediated proliferation. This logic extends all the way through the in vivo experiments (which are quite promising, translationally).

Our data demonstrated that Gli1^+^ cells possess the properties of mesenchymal progenitors: self-renewal and multi-lineage differentiation. Since CFU-Fs formed from digested *Gli1ER/Td* meniscus cells were 73% Td^+^ and 27% Td^-^, we believe that the major portion of meniscal mesenchymal progenitors are Gli1^+^ cells. Sca1, CD90, CD200, and PDGFR were selected as mesenchymal progenitor markers according to studies performed in other tissues, such as bone marrow. No thorough investigation of these markers has been previously conducted in the meniscus tissue. Our data showed that Sca1^+^, CD90^+^, CD200^+^, and PDGFR^+^ cells are 25, 11, 41, and 19% of digested meniscus cells, respectively. Considering CFU-F frequency of digested meniscal cells is ~1%, these markers must label non-progenitors as well. To be more precise, we changed the title of Results subsection to “Gli1-expressing meniscus cells are a major subset of mesenchymal progenitors”.

4. The spatial profile of Gli1-expressing cells in the meniscus is beautifully described, however an interpretation for the superficially restricted zonation of Gli1 reporter activity is not given. Do these superficial cells have more or less cartilage antigen expression? Is there something clearly physiologically different in the Gli1-rich superficial layers that could be determined? Line 401 cites an osteoblast paper to set up the relevance of Gli1+ cells in development of musculoskeletal tissues. However, the meniscus is much more similar to the enthesis and the PDL. The authors should therefore lead with that literature. The PDL literature in particular is not cited and should be added. Also missing are recent enthesis development/regeneration papers (PMID: 30504126, 26141957, and 28219952).

As shown in Figure 2A, meniscal superficial layer is labeled by Prg4, a lubricant protein. We found that Gli1^+^ cells are also Prg4^+^. The Rosen group previously characterized the zones of mouse meniscus ^(2)^. At 2 weeks of age, outer region ﬁbroblast-like cells are type I collagen positive, while inner region ﬁbro-chondrocytes express type II collagen and surround an area of larger, round cells that resemble hypertrophic chondrocytes and transiently express type X collagen. At 8 weeks of age, Safranin O proteoglycan staining is concentrated in the superﬁcial zone surrounding the ﬁbro-chondrocytes. Our study showed a similar staining pattern (Figure 4). We thank the reviewers for the reference suggestions. We added them to our discussion accordingly.

5. The characterization of Gli1+ and Gli1- FAC sorted cells could be expanded on a bit.

Characterization of freshly sorted Gli1^+^ and Gli1^-^ cells is technically challenging. Due to low abundance of Gli1^+^ cells and small size of mouse meniscus, at least 15 mice are required for collecting 10,000 Gli1^+^ cells for qRT-PCR analysis. Thus, we decided to expand cells in culture first and then performed proliferation (Figure 2—figure supplement 2), migration, and multi-lineage differentiation assays (Figure 2).

6. CFU-F images should be provide in addition to quantification. The differentiation studies in Figure 2E are non-quantitative and not convincing. Further, it is a little contradictory that under certain contexts Gli1+ cells form more cartilage (2E), but under other culture conditions they have reduced cartilage markers (2F). These points need to be clarified.

We seeded total digested cells for CFU-F assay and counted Td^+^ and Td^-^ CFU-Fs within the same dish (Author response image 1). Thus, we do not think it is necessary to show the dish or CFU-F colony images. We performed qRT-PCR analysis of lineage specific markers in Gli1^+^ and Gli1^-^ cells after multi-lineage differentiation. Quantification confirms that Gli1^+^ cells have better differentiation ability than Gli1^-^ cells. These data are now included in Figure 2.

**Author response image 1. sa2fig1:** CFU-F assay of total digested meniscus cells. (A) Crystal-violet staining of CFU-F colonies in a dish. (B) Brightfield and fluorescence images of a representative Gli1^+^ CFU-F colony in the dish.

7. In Figure 5, changes in distribution or survival of Gli1+/- cells may underlie the difference, but survival nor Gli1- cell distribution were not assessed.

TUNEL staining of meniscus at 2, 4 and 8 weeks after injury did not detect any TUNEL^+^ Gli1^+^ (yellow) cells (Author response image 2). However, we did observe some TUNEL^+^ Gli1^-^ (green) cells inside the meniscus. Since Gli1^+^ cells are mostly located at the meniscus surface, we believe that they are cleared by macrophage in a more rapid way than cells inside the meniscus.

**Author response image 2. sa2fig2:** Representative TUNEL fluorescent staining of *Gli1ER/Td* mouse knee joints harvested at 2, 4, or 8 weeks after meniscus injury. Mice received Tamoxifen injections right before the injury. Boxed area at the top is shown at a high magnification at the bottom (Scale bars, 200 μm). F: femur; T: tibia; MS: meniscus synovial end; ML: meniscus ligamental end. Red: Td; Green: TUNEL; Blue: DAPI.

8. Cartilage differentiation within the meniscus appears to be promoted with Gli1+ cell therapy and Purmorphamine. This could be assessed. Similarly, Hh signaling is known to induce osteogenesis. Osteoblastic antigens and/or presence of osteophytes should be assessed for in purmorphamine treated joints.

We agree that Gli1^+^ cell therapy and Purmorphamine (PMA) treatment appear to increase the Safranin O staining in meniscus after injury (Figure 5). These results are consistent with our data that Gli1^+^ meniscus cells differentiate into chondrocytes better than Gli1^-^ meniscus cells (Figure 2) and reports from other groups that Hh signaling activation promotes chondrogenic differentiation in bone marrow mesenchymal progenitors ^(5,6)^. However, since this increase of staining is relatively subtle, we are unable to further assess it by qRT-PCR, which, due to technical difficulties, analyzes the entire meniscus instead of the injury site.

To address the issue whether PMA stimulates ectopic ossification, we used microCT to analyze joint calcification at 3 months post meniscal injury with or without PMA treatment. We detected no change in the calcified meniscus volume and a trend of decrease in osteophyte volume in PMA-treated mice compared to vehicle-treated mice. Thus, our data, now included as Figure 5—figure supplement 1, suggest that Hh agonist does not affect overall joint calcification.

9. One topic that is not covered in the paper is the role of Hh signaling in chondrocyte mineralization. This has been well studied in the growth plate (esp. related to PTHrP / IHH feedback loop) and may have relevance to the meniscus as well. The healing studies should consider this carefully, as ectopic mineralization is a possible negative side effect of Hh treatment.

We agree that this is a valid concern for stimulating the Hh pathway in the joint. As explained in our answer to the previous comment, we did not observe obvious changes in cartilage differentiation and tissue mineralization after PMA treatment. Our PMA treatment regimen is only a single injection, which likely mitigates effects that could arise from prolonged over-expression of the pathway, as seen in other genetic models ^(7)^. Although we do not see evidence supporting this in our mouse model, we will keep this potential side effect in mind when we proceed to large animal models for future translational studies. We thank the reviewers for raising this clinically important question.

10. There are a number of places in the results where it is unclear if the authors are talking about Gli+ cells or Gli1-lineage cells. This should be clarified throughout, perhaps with specific nomenclature that defines "Gli1+" as cells that are positive for Gli and "Gli1-lineage" for cells that are descendants of Gli+ cells. Supplemental Figure 1A should be in the main document. Similar schematics in other figures are very useful for understanding the experiment.

We thank the Reviewers for this suggestion that greatly improves the accuracy of our manuscript. We now use Gli1^+^ cells for Td^+^ cells right after Tamoxifen induction and Gli1 lineage cells for Td^+^ cells in lineage tracing experiments.

11. What are the temporal expression patterns of Gli1 and other Hh related genes during development and healing? It would be informative to see localized expression (e.g., in situ hybridization) or qPCR expression for healing tissues.

Please see our reply to comment #2.

12. The authors should clarify a number of things with meniscal cell isolation: (a) There are clearly differences in cell phenotype between superficial and deep areas and between attachment and midsection; was this considered for cell isolation? (b) I assume TAM injections were performed and then cells were isolated a few days later via FACS; please clarify details to show that Gli1+ (not Gli1-lineage) cells were characterized. (c) Figure 2: 3-month old mice were used, but again, Gli+ vs. Gli1-lineage cells is not indicated.

We apologize for not stating these details in the first submission. For Figure 2, we injected 3-month-old *Gli1ER/Td* mice with Tam for 5 consecutive days and euthanized them 2 days after the last injection for immunostaining and cell isolation experiments. We isolated the meniscus cells from the entire meniscus tissue. It is quite difficult, if not impossible, to cleanly dissect different regions of meniscus even under dissection microscopy. Since we analyze Td^+^ cells right after Tam induction, those cells are Gli1^+^ cells not Gli1-lineage cells. We now add those details to Methods and Results. In the future, we will harvest meniscus cells from different regions of porcine or bovine meniscus to address this issue.

13. The mechanisms by which Gli1+ and Hh treatments work is not explored. Some of the results are counter-intuitive. For example, why would Hh stimulate proliferation if Gli1+ cells if these are thought to be slow turnover resident stem cells? Furthermore, why would Hh stimulation lead to proliferation rather than differentiation, (in contrast to what is know in growth plate biology)?

Slow cycling cells, also called label-retaining cells (LRCs), are considered tissue stem cells. Those cells might change their cycling frequency depending on the environment (niches) and tissue status (development, homeostasis, or injury) ^(8)^. Hh signaling is known to regulate proliferation and differentiation of stem/progenitor cells in musculoskeletal tissues. For example, during endochondral ossification and growth plate development, Hh signaling stimulates chondrocyte proliferation and inhibit chondrocyte hypertrophy indirectly via regulating PthrP ^(9)^. In bone, Hh signaling stimulates both proliferation and differentiation of mesenchymal progenitors ^(10)^. In enthesis, Hh signaling maintains Gli1^+^ cells and regulates Gli1^+^ progenitor proliferation and differentiation ^(11-14)^. In our studies, we observed that in vivo PMA treatment increases Gli1-lineage cells after injury (Figure 5H) and that in vitro PMA treatment stimulates proliferation and migration of meniscal mesenchymal progenitors (Figure 3). We do not know whether PMA treatment directly stimulates meniscus differentiation due to a lack of meniscus specific markers. Thus, here we focus more on proliferation, rather than differentiation.

14. The assessment of healing is qualitative/semi-quantitative (histomorphometry). The authors should perform a more rigorous assessment of healing to demonstrate the effectiveness of the Gli1+ cell and Hh therapies. This should include quantitative outcome(s) such as qPCR, mechanics, etc.

The healing was mostly evaluated by histological observation of bridging at the injury site. To our knowledge, no genes have been commonly agreed upon to represent healed meniscus. Thus, we are unable to perform qRT-PCR analysis. However, since meniscus injury causes osteoarthritis and joint pain is commonly used as a functional outcome of osteoarthritis, we used Mankin Score (for cartilage degeneration) and von Frey assay (for joint pain) to indirectly quantify the healing outcomes (Figure 6).

We agree with the reviewer that the biomechanical properties of healing meniscus are an important parameter of healing. In the past, we have applied AFM-nanoindentation to quantify the mechanical properties of healthy, adult meniscus central region ^(15)^. However, limited by the small size and irregular shape of murine meniscus, there is no available method for directly assessing the biomechanical properties of the healing meniscus site at the horn region. Our technique is not readily applicable for this study yet. Further tuning in sample embedding, preparation/sectioning, and healing site identification are needed to apply the AFM method, which are not trivial. In the future, we will definitely develop a suitable approach to measure the biomechanical properties of healing mouse meniscus.

15. The Gli1+ cell therapy histologic results are impressive. This is surprising because the delivery method was relatively simple. How much cell engraftment was there? Can the authors comment further (or add experiments to elucidate) on how long the cells were present and what their direct involvement was in healing?

As shown in Author response image 3, after Td(Gli1)^+^ cells injection, Td^+^ cells peaked around 4 weeks and then declined at 8 weeks. We believe that injected Gli1^+^ cells contribute to the healing process in multiple ways. First, they directly differentiate into meniscal cells. Second, they indirectly secrete factors regulating endogenous cells to accelerate healing. Third, their high migration ability facilitates the bridging of injury site. Indeed, the mechanisms by which MSCs contribute to tissue regeneration has been extensively studied for many years but still remain largely unclear ^(16)^. Therefore, to investigate the detailed mechanism is very interesting but beyond the scope of our current research.

**Author response image 3. sa2fig3:** Representative confocal images of *WT* mouse knee joints at 1, 4, and 8 weeks after meniscus injury and injection of Gli1^+^ cells derived from *Gli1ER/Td* meniscus. Boxed areas in the top panel are shown at high magnification at the bottom. Dashed line outlines meniscus. Scale bars, 200 μm. Blue: DAPI; Red: Td.

16. The authors show that native Gli1+ cells expand after injury. If this is the case, what is the rationale for adding more Gli1+ cells? Is the idea that the tissue has the capacity to heal but there aren't enough native Gli1+ cells to do the job?

We agree with the Reviewers that meniscus tissue might not have enough native Gli1^+^ cells after injury to do the healing job. Therefore, addition stimulation is required to accelerate healing. This seems to be a common theme in degenerative diseases. Another project in our group studies whether EGFR pathway can be targeted for OA treatment ^(17)^. We and others have shown that articular cartilage expresses EGFR and its ligands. While EGFR activity decreases at early OA stage, it reappears during middle OA stage. However, the endogenous EGFR activity is apparently not enough to protect cartilage from OA damage. Genetically overexpression EGFR ligand or administration of EGFR ligand-conjugated nanoparticles prevents OA progression in mouse OA models.

17. Figures and text jump between methodologies, making interpretation of results difficult. Figure 1 shows that superficial cells of the meniscus generally have active Hh signaling 24-hours prior to a variety of postnatal-to-adult timepoints (A, B, E, F), and postnatal Hh signaling drives proliferation of early meniscus cells (C, D). It does not appear to report any long-term pulse/chase lineage tracing experiment as suggested in the text (Lines 223+). If this interpretation is incorrect, perhaps this could be addressed by increased clarity of figures and text (Methods, Results, Figure organization and captions)?

The long term chasing images are shown in the bottom row of Figure 1A (d,h,l,p,t).

References:

1. Fabian SL, Penchev RR, St-Jacques B, et al. Hedgehog-Gli pathway activation during kidney fibrosis. Am J Pathol. 2012;180(4):1441-53.

2. Gamer LW, Xiang L, Rosen V. Formation and maturation of the murine meniscus. J Orthop Res. 2017;35(8):1683-9.

3. Baker BM, Nathan AS, Huffman GR, Mauck RL. Tissue engineering with meniscus cells derived from surgical debris. Osteoarthritis Cartilage. 2009;17(3):336-45.

4. Han WM, Heo SJ, Driscoll TP, et al. Microstructural heterogeneity directs micromechanics and mechanobiology in native and engineered fibrocartilage. Nat Mater. 2016;15(4):477-84.

5. Handorf AM, Chamberlain CS, Li WJ. Endogenously produced Indian Hedgehog regulates TGFb-driven chondrogenesis of human bone marrow stromal/stem cells. Stem Cells Dev. 2015;24(8):995-1007.

6. Steinert AF, Weissenberger M, Kunz M, et al. Indian hedgehog gene transfer is a chondrogenic inducer of human mesenchymal stem cells. Arthritis Res Ther. 2012;14(4):R168.

7. Lin AC, Seeto BL, Bartoszko JM, et al. Modulating hedgehog signaling can attenuate the severity of osteoarthritis. Nature Medicine. 2009;15(12):1421-5.

8. Fuchs E. The tortoise and the hair: slow-cycling cells in the stem cell race. Cell. 2009;137(5):811-9.

9. Ohba S. Hedgehog Signaling in Skeletal Development: Roles of Indian Hedgehog and the Mode of Its Action. Int J Mol Sci. 2020;21(18):6665.

10. Shi Y, He G, Lee WC, McKenzie JA, Silva MJ, Long F. Gli1 identifies osteogenic progenitors for bone formation and fracture repair. Nat Commun. 2017;8(1):2043.

11. Schwartz AG, Long F, Thomopoulos S. Enthesis fibrocartilage cells originate from a population of Hedgehog-responsive cells modulated by the loading environment. Development. 2015;142(1):196-206.

12. Schwartz AG, Galatz LM, Thomopoulos S. Enthesis regeneration: a role for Gli1+ progenitor cells. Development. 2017;144(7):1159-64.

13. Felsenthal N, Rubin S, Stern T, et al. Development of migrating tendon-bone attachments involves replacement of progenitor populations. Development. 2018;145(24):dev165381.

14. Dyment NA, Breidenbach AP, Schwartz AG, et al. Gdf5 progenitors give rise to fibrocartilage cells that mineralize via hedgehog signaling to form the zonal enthesis. Dev Biol. 2015;405(1):96-107.

15. Li Q, Doyran B, Gamer LW, et al. Biomechanical properties of murine meniscus surface via AFM-based nanoindentation. J Biomech. 2015;48(8):1364-70.

16. Sagaradze GD, Basalova NA, Efimenko AY, Tkachuk VA. Mesenchymal Stromal Cells as Critical Contributors to Tissue Regeneration. Front Cell Dev Biol. 2020;8:576176.

17. Wei Y, Luo L, Gui T, et al. Targeting cartilage EGFR pathway for osteoarthritis treatment.. Sci Transl Med 2021; 13(576):eabb3946.